# The Influence of Dietary Patterns on Polycystic Ovary Syndrome Management in Women: A Review of Randomized Controlled Trials with and Without an Isocaloric Dietary Design

**DOI:** 10.3390/nu17040674

**Published:** 2025-02-13

**Authors:** Yeonjeong Choi, Kyeonghoon Kang, Minkyung Je, Young-Min Lee, Yoona Kim

**Affiliations:** 1Department of Food and Nutrition, Gyeongsang National University, Jinju 52828, Republic of Korea; duswjd6638@gnu.ac.kr (Y.C.); pos26109@gmail.com (K.K.); alsrud4687@naver.com (M.J.); 2Department of Practical Science Education, Gyeongin National University of Education, Gyesan-ro 62, Gyeyang-gu, Incheon 21044, Republic of Korea; ymlee@ginue.ac.kr; 3Department of Food and Nutrition, Institute of Agriculture and Life Science, Gyeongsang National University, Jinju 52828, Republic of Korea

**Keywords:** polycystic ovary syndrome, dietary pattern, randomized controlled trial

## Abstract

Polycystic ovary syndrome (PCOS) is an endocrine disorder that causes cardiometabolic and reproductive disorders in women of reproductive age. Women with PCOS are more likely to have obesity, type 2 diabetes mellitus, and cardiovascular disease. There is an inconclusive consensus on which dietary modification could be most effective in PCOS prevention and treatment. This review aimed to examine the effects of diverse dietary patterns on PCOS in women according to randomized controlled trials (RCTs) with and without an isocaloric dietary design. A literature search was performed in the PubMed^®®^/MEDLINE^®®^ database up to 14 November 2024. A total of 21 RCTs were reviewed after screening the records, including 15 RCTs with a calorie-restricted dietary design and 6 RCTs with a non-calorie-restricted dietary design. This review found beneficial effects of the calorie-restricted Dietary Approaches to Stop Hypertension (DASH) diet on weight loss and glucose control in women with PCOS in four RCTs with an isocaloric dietary design. The calorie-restricted low-glycemic index (GI) diets from three RCTs and high-protein diets from four RCTs with an isocaloric dietary design showed no significant differences in anthropometric parameters, glucose control, lipids, and gonadal parameters compared with the control diet in women with PCOS. Non-calorie-restricted low-carbohydrate diets from four RCTs with an isocaloric dietary design showed similar results to the calorie-restricted low-GI diets and high-protein diets. However, the existing number of RCTs is insufficient to conclude the association between dietary patterns and PCOS in women. Further, well-designed dietary intervention studies are needed to assess the role of dietary patterns in PCOS beyond calorie restriction.

## 1. Introduction

Polycystic ovary syndrome (PCOS) is a complex, multifactorial endocrine disorder that is the most prevalent in women of reproductive age [1]. PCOS is diagnosed by the presence of at least two of the following based on the Rotterdam criteria: ovarian cysts assessed by ultrasound examination, clinical hyperandrogenism with high circulating androgen levels, and oligo-amenorrhea with oligo-anovulation [2]. The most common symptoms observed include hirsutism, alopecia, and acne, along with oligomenorrhea and amenorrhea, which are linked to the diagnosis criteria [3]. As PCOS is also associated with an increased risk of type 2 diabetes mellitus (T2DM), obesity, and cardiovascular disease complications [4], it is crucial to prevent and manage PCOS effectively.

Treatment and management for PCOS include lifestyle modification, medications, balanced diets, and weight loss through regular physical activity [5,6,7,8,9,10]. An appropriate diet with well-balanced nutrients is a safe and long-term non-pharmacological strategy for PCOS prevention and treatment [10,11,12]. Scientists are especially actively working to identify a diet that could support the prevention and management of PCOS. A meta-analysis of 19 Randomized Controlled Trials (RCTs) [13,14,15,16,17,18,19,20,21,22,23,24,25,26,27,28,29,30,31] to determine whether diet improves insulin sensitivity (Si) in women with PCOS indicated that dietary intervention could be beneficial in the management of PCOS [32]. Nineteen RCTs evaluated 10 low-carbohydrate diets [13,14,15,16,17,18,22,26,27,31], 4 Dietary Approaches to Stop Hypertension (DASH) diets [19,20,23,24], 3 calorie-restricted diets [21,28,29], a low-fat diet [30], and a Mediterranean (MED) diet [25]. Dietary intervention periods included in this meta-analysis ranged from 4 weeks to 1 year. Dietary interventions significantly attenuated fasting blood glucose (FBG), fasting blood insulin (FBI), and Homeostasis Model Assessment of Insulin Resistance (HOMA-IR) in a meta-analysis of 5 [16,18,20,24,27], 9 [13,15,16,18,20,22,24,26,27] and 6 [15,17,18,20,24,27] RCTs, respectively.

The dietary intervention significantly reduced body mass index (BMI), body weight (BW), and waist circumference (WC) from 9 [15,17,19,20,21,22,24,25,27], 12 [13,14,15,17,19,20,21,23,24,25,26,27], and 5 [14,17,21,23,27] RCTs, respectively. A positive correlation between PCOS and BW has been observed. Obesity, particularly visceral adiposity common in women with PCOS, appears to exacerbate all reproductive and metabolic symptoms in individuals with PCOS [33]. Weight loss of as little as 5 to 10% can attenuate androgenemia menstrual irregularities and infertility cardiometabolic risk in women with PCOS [5,10,34]. A recent systematic review and meta-analysis (15,129 women) reported that the increased prevalence of overweight and obesity in individuals with PCOS was estimated to be 1.95 (95% confidence interval (CI) 1.52 to 2.50), 2.77 (95% CI = 1.88 to 4.10) compared with individuals without PCOS, respectively [35]. However, as PCOS is prevalent and cardiovascular risk factors are prevalent in women with PCOS across BW [36], there is still no consensus on which weight loss-induced dietary modifications can be better in PCOS management.

Higher consumption of foods high in calories, saturated fat, and glycemic index (GI) and low in dietary fiber was observed in women with PCOS compared with controls [37,38,39,40], which can elevate chronic low-grade inflammation and deteriorate metabolic and reproductive function related with the pathophysiology of PCOS [41,42,43,44,45,46].

However, there is still no consensus on which dietary modifications can be better in PCOS management. To address the research question, this review investigated the impact of various dietary patterns on women with PCOS based on RCTs with an isocaloric dietary design and without an isocaloric dietary design.

## 2. Materials and Methods

### 2.1. Search Strategy

A flow chart of the study’s screening and selection process is reported in Figure 1. This study investigated the influence of dietary patterns on PCOS in women in RCTs. The literature research was performed using the PubMed^®®^/MEDLINE^®®^ (https://pubmed.ncbi.nlm.nih.gov/pubmed/) database up to 14 November 2024. The search strategy consisted of RCTs combined with dietary patterns, including but not limited to calorie-restricted diet or DASH diet or MED diet or ketogenic diet (KD) or low-carbohydrate diet or high-protein diet or low-fat diet or low-GI diet or low-glycemic load (GL) diet or plant-based diet or vegetarian diet or vegan diet or healthy eating index or alternative healthy eating index or Western diet or and other relevant terms. In addition, reference lists of studies were searched manually to find any relevant publications.

### 2.2. Inclusion and Exclusion Criteria

Two investigators independently screened the title and/or abstract of studies, with differences consented upon the presence of a third investigator. The remaining studies were reviewed with the full texts of literature for the final study selection. Studies were included if they met the following criteria: RCT or randomized crossover study, full-text available, and assessing the effect of different dietary patterns on PCOS in women (anthropometric parameters, glucose control, lipids, gonadal parameters, and so on).

Exclusion criteria were as follows: non-RCTs, including observational studies, reviews, comments, editorials, and case reports; studies with non-human studies; pre-post studies; studies with unrelated intervention design; studies that included irrelevant study subjects; and full-texts not available in English language. With regard to the unrelated intervention design, this study excluded studies if the intervention included physical exercises, medications, extracts, or supplements. This study excluded studies that included subjects without PCOS and studies that assessed outcomes not outlined above. Further exclusion was performed in studies that included subjects who were not adults or had a history of metformin usage.

## 3. Results

This review identified a total of 282 studies, which included 273 from a database search and 9 from a manual search. Of these, 244 studies were initially excluded: 161 were excluded due to their publication type and 83 were excluded for having unrelated intervention designs. Furthermore, 17 studies were excluded through full-text review, 15 studies were excluded due to their inclusion of irrelevant study subjects, and 2 studies with non-English full texts were excluded. Finally, 21 studies were included in this review (Figure 1).

A total of 21 RCTs [13,14,15,16,17,18,19,20,23,24,47,48,49,50,51,52,53,54,55,56,57] that meet the inclusion criteria were selected and included in this review. The total number of involved subjects is 922 PCOS women aged 18 to 51 years. The study periods ranged from 4 weeks to 6 months. The 17 RCTs [13,14,15,16,19,20,23,24,47,48,50,51,52,53,54,56,57] recruited overweight and/or obese women, while the BMI range was not clear in 4 RCTs [17,18,49,55]. In terms of a study design, 16 RCTs [13,14,15,16,17,19,20,23,24,47,48,49,50,51,52,53] are parallel designs, while 5 RCTs are crossover designs [18,54,55,56,57]. The RCTs were conducted in Iran [16,19,20,23,24], China [51], Mexico [49], USA [14,18,52,54,55,56,57], Italy [47], Australia [13,15,48], UK [50,53], and Denmark [17]. A total of 15 [13,14,15,16,17,18,19,20,23,24,47,48,49,50,51,52,53,54,55,56,57] of 21 RCTs [13,14,15,16,17,18,19,20,23,24,47,48,49,50,51,52,53,54,55,56,57] restricted the total daily energy intake, while 6 RCTs [17,18,54,55,56,57] did not restrict the total daily energy intake. Table 1 summarizes the effects of dietary patterns in individuals with PCOS from isocaloric dietary RCTs [13,14,15,16,17,18,19,20,23,24,48,49,50,52,54,55,56,57], while Table 2 highlights the effects of dietary patterns from dietary RCTs with differing calorie levels [47,51,53].

### 3.1. Calorie-Restricted Diet

A calorie-restricted diet is a diet that reduces sustainable energy intake compared with the amount of energy required to maintain BW while providing enough energy to maintain metabolic balance. It is essential to supply sufficient amounts of micronutrients and fiber to maintain a high-quality diet [58]. A calorie-restricted diet is associated with weight loss, fat mass reduction, higher Si, and anti-oxidative stress [59].

A total of 15 RCTs [13,14,15,16,19,20,23,24,47,48,49,50,51,52,53] that conducted calorie-restricted dietary interventions were included in this review. A total of 12 [13,14,15,16,19,20,23,24,48,49,50,52] of 15 RCTs [13,14,15,16,19,20,23,24,47,48,49,50,51,52,53] were a dietary intervention design with an isocaloric match, while 3 RCTs [47,51,53] were not an isocaloric-matched intervention design. The dietary patterns of 15 RCTs [13,14,15,16,19,20,23,24,47,48,49,50,51,52,53] with a calorie-restricted intervention design are as follows: DASH diet [19,20,23,24], KD [47], low-carbohydrate diet [48], low-GI diet [16,49,50], high-protein diet [13,14,15,51,52], and low-calorie diet [53].

#### 3.1.1. DASH Diet

DASH is a dietary pattern that promotes the intake of fruits, vegetables, whole grains, low-fat dairy products, poultry, fish, and nuts. The DASH dietary pattern is rich in protective nutrients such as potassium, calcium, magnesium, and fiber while limiting the intake of refined carbohydrates, total fat, saturated fat, and cholesterol [60,61]. The DASH diet has been shown to significantly lower both systolic blood pressure (SBP) and diastolic blood pressure (DBP) in individuals with hypertension and has also been associated with improvements in insulin resistance, inflammation, oxidative stress, and other cardiovascular risk factors, such as triglyceride (TGs), high-density lipoprotein cholesterol (HDL-C), FBG, and TC levels [60,62,63,64].

In this review, four RCTs [19,20,23,24] examined the effect of a calorie-restricted DASH diet on PCOS in overweight and/or obese women compared with an isocaloric control diet. The four RCTs [19,20,23,24] conducted dietary interventions for a period of 8 weeks to 12 weeks in 211 subjects aged between 18 and 40 years.

In a randomized placebo-controlled parallel study by Azadi-Yazdi et al. 2017 [23] of overweight and/or obese women with PCOS whose ages ranged from 20 to 40 years, adherence to the calorie-restricted DASH diet [intervention group (InG); *n* = 28] for 12 weeks significantly lowered BW, BMI, total fat mass (TFM), and androstenedione (A4) and increased sex hormone binding globulin (SHBG) compared with the calorie-restricted control diet [control group (CG); *n* = 27]. No changes in WC, hip circumference (HC), lean body mass (LBM), waist-to-hip ratio (WHR), total testosterone (T), and free androgen index (FAI) were observed. The InG and CG were designed to match calories with the same nutrient composition (50 to 55% carbohydrate, 15 to 20% protein, and 25 to 30% fat from total daily energy intake). Women in the InG were instructed to increase their intake of whole grains, nuts, seeds and legumes, vegetables, and fruits and decrease their intake of grains and simple sugar. In contrast, women in the CG were instructed to increase grains and simple sugar [23] (Table 1).

In a randomized parallel trial by Foroozanfard et al., 2017 [24], overweight and/or obese women with PCOS who consumed a calorie-restricted DASH dietary pattern (InG; *n* = 30) for 12 weeks showed significant decreases in BW and BMI compared with the calorie-restricted control diet (CG; *n* = 30). The InG significantly decreased FBI, HOMA-IR, and homeostasis model assessment of beta-cell function (HOMA-B), while the InG increased the quantitative insulin sensitivity check index (QUICKI) compared with the CG. However, no change in FBG was found between the InG and the CG. In addition, in the InG there was lowered anti-Müllerian hormone and FAI and elevated SHBG compared with the CG. No differences in total T, follicle-stimulating hormone (FSH), luteinizing hormone (LH), and 17-hydroxyprogesterone were found between the InG and the CG. Moreover, in the InG there was an increased nitric oxide level and decreased malondialdehyde (MDA) level compared with the CG. The InG and the CG were isocaloric matched and were the same in macronutrient composition (52 to 55% carbohydrate, 16 to 18% protein, and 30% fat from total daily energy intake). Women who followed in the InG were advised to increase the intakes of whole grains (at least 3 servings/day), vegetables, fruits, low-fat dairy (<2%), nuts, seeds, and legumes; moderate in meats, poultry, and fish. In contrast, women in the CG were advised to increase their intake of grains and simple sugar and lower their intake of dairy, nuts, seeds and legumes, vegetables, and fruits [24] (Table 1).

Asemi et al., 2015 [20] conducted a randomized parallel trial to examine the effect of the DASH diet on anthropometric parameters and glucose controls in overweight and/or obese women with PCOS. The women with PCOS were randomized to either the calorie-restricted DASH diet (InG; *n* = 24; 52% carbohydrate, 18% protein, and 30% fat from total daily energy intake; sodium < 2400 mg/day) or the control diet (CG; *n* = 24), which matched the macronutrients composition of the DASH diet. The 1800 kcal DASH diet consisted of nine servings of grains (at least three from whole grains), two servings of simple sugar, five servings of vegetables, six servings of fruits, three servings of low-fat dairy, four servings of lean meat, poultry, and fish, two servings of nuts and legumes, and three servings of fats and oils. In contrast, women who in the CG consumed higher servings of simple sugar, meat, poultry, and fish while consuming lower servings of vegetables, fruits, nuts, seeds, and legumes compared with those following the DASH diet. After 8 weeks of dietary intervention, women in the InG had lower anthropometric parameters (BW, BMI, WC, and HC), glucose control (FBI and HOMA-IR), and QUICKI compared with the CG. Conversely, there were no significant differences in FBG and HOMA-B between the InG and the CG [20] (Table 1).

In addition, Asemi et al., 2014 [19] assessed the effect of adherence to a calorie-restricted DASH diet on lipids with the same intervention design as the previous RCT [20]. They found no significant changes in TC, TG, very low-density lipoprotein (VLDL-C), HDL-C, LDL-C, and TC/HDL-C ratio [19] (Table 1).

In summary of this review, the four RCTs [19,20,23,24] investigated the effect of calorie-restricted DASH diet on PCOS in overweight and/or obese women with an isocaloric dietary design. With regard to anthropometric parameters, the DASH diet showed beneficial effects on BW [19,20,23,24] and BMI [19,20,23,24] compared with the control diet. One [20] of two RCTs [20,23] showed significant decreases in WC and HC. The DASH diet reduced FBI [20,24] and HOMA-IR [20,24] while increasing QUICKI [20,24], but no significant difference in FBG between the DASH diet and the control diet was found [20,24]. These indicate that the DASH diet has a positive effect on glucose control. One RCT [19] showed no significant changes in TC, LDL-C, HDL-C, VLDL-C, TG, and TC/HDL-C ratio, indicating additional interventions to determine the effect of the DASH diet on lipids in women with PCOS. In two RCTs [23,24] with an isocaloric dietary design, the DASH diet was found to increase the level of SHBG compared with the control diet, while total T showed no significance. In conclusion, the DASH diet appeared to be effective in BW and glucose control in women with PCOS using an isocaloric dietary design.

#### 3.1.2. Ketogenic Diet

The KD, characterized by high fat, low carbohydrates, and adequate protein, mimics a state of the metabolic state of fasting in which ketone bodies derived from fats become the primary energy source [65]. Ketone bodies serve as an alternative energy source to glucose and are also crucial for brain development, synthesizing cell membranes and lipids [66]. The KD has primarily been used to treat neurological disorders, especially epilepsy; however, it has also shown benefits in weight loss, cardiovascular disease, and T2DM [67].

In this review, only one RCT [47] investigated the effect of a calorie-restricted KD on PCOS in women with a non-isocaloric control diet.

An RCT conducted by Cincione et al., 2023 [47] recruited overweight and/or obese women with PCOS aged 18 to 45 years. The adherence to a calorie-restricted KD (InG; *n* = 73) for 45 days significantly decreased anthropometric parameters (BW, BMI, WC, HC, WHR, TFM, fat-free mass (FFM), basal metabolism, and total body water), glucose control (FBG, FBI, HOMA-IR, and C-peptide), and gonadal parameters (total T, free T, and LH) compared with the control calorie-restricted MED diet (CG; *n* = 71). In addition, the InG significantly increased FSH and SHBG compared with the CG. Women who followed the InG were advised to limit carbohydrates (≤30 g per day), maintain proteins (1.1 to 1.2 g protein/ideal BW), and increase lipids consumption (30 g/day; 10 g from EVOO), while women who followed the CG were advised to increase intake of whole grains. The InG and the CG were not isocaloric matched [47] (Table 2).

#### 3.1.3. Low-Carbohydrate Diet

A low-carbohydrate diet is defined as a dietary pattern that limits the consumption of carbohydrates to a proportion of total daily energy intake by 20% [68] or 10 to 25% [69]. A low-carbohydrate diet (<26% carbohydrate from total daily energy intake) showed higher reductions in BW and fat mass compared with the MED diet in overweight and obese adults [70]. In addition, a low-carbohydrate diet was associated with decreases in BMI, hemoglobin A1c (HbA1c), and FBG in patients with T2DM [71].

In this review, only one RCT [48] examined the effect of a calorie-restricted low-carbohydrate diet on the anthropometric parameters, glucose control, and gonadal parameters in women with PCOS.

Moran et al., 2006 [48] randomly allocated overweight and/or obese women with PCOS to either a calorie-restricted low-carbohydrate diet (InG) or the control diet (CG). In the first 8 weeks, the InG and the CG were instructed to consume the same three meals per day: two meals with replacements (each meal: ~430 kcal), and a low-fat evening meal (~836 kcal; fruit and vegetables: ≥5 servings/day). The InG was instructed to restrict carbohydrate consumption to less than 120 g per day, while the isocaloric-matched CG was instructed to restrict fat consumption to less than 50 g per day for 8 to 32 weeks. After the first 8 weeks, a calorie-restricted diet in both the InG and CG significantly decreased anthropometric parameters (BW, WC, TFM, FFM, SBP, resting energy expenditure, and respiratory quotient), glucose control (FBG, FBI, and HOMA-IR), and gonadal parameters (total T, free T, and FAI) in 34 women compared with the baseline. In addition, a significant increase in SHBG was found in the InG and the CG compared with the baseline, while no differences were found in DBP and glucose area under the curve (AUC). However, no significant differences were found in the anthropometric parameters (BW, WC, TFM, FFM, SBP, DBP, resting energy expenditure, and respiratory quotient), glucose control (FBG, FBI, HOMA-IR, and glucose AUC), and gonadal parameters (total T, free T, FAI, and SHBG) between the InG and the CG from 8 to 32 weeks [48] (Table 1).

#### 3.1.4. Low-GI Diet

The GI is the incremental area under the blood glucose curve following the ingestion of carbohydrates in a food, expressed as a percentage of the corresponding area following the same amount of carbohydrates from a standard food, either glucose or white wheat bread [72]. The low-GI foods (≤70) are considered to be favorable in terms of health, particularly for the prevention of obesity, T2DM, and cardiovascular disease [73]. A low-GI diet has been associated with reductions in HbA1c, FBG, BMI, TC, and LDL-C in individuals with type 1 diabetes, T2DM, or impaired glucose tolerance [74].

In this review, the three RCTs [16,49,50] investigated the effect of a calorie-restricted low-GI diet on anthropometric parameters, glucose control, lipids, and gonadal parameters in women with PCOS. Two [49,50] of three RCTs [16,49,50] examined the effect of a calorie-restricted low-GI diet on PCOS in women with the control diet. One [16] of the three RCTs [16,49,50] examined the effect of a calorie-restricted low-GI plus high-protein diet on PCOS in women compared with the control diet.

Sordia-Hernández et al., 2016 [49] conducted a randomized parallel study to examine the effect of the low-GI diet on ovulation compared with the control diet for 3 months. A total of 37 women with PCOS were advised to receive either the calorie-restricted low-GI diet (InG; *n* = 19; 45% to 50% complex carbohydrate, 15% to 20% protein, and 30% to 40% fat—10% to 15% monounsaturated fatty acid (MUFA), <10% polyunsaturated fatty acid (PUFA), and <10% saturated fatty acid (SFA)—from total daily energy intake; 20 to 35 g fiber per day; <45 GI) or the isocaloric control diet (CG; *n* = 18; 45% and 50% complex carbohydrate, 15% to 20% protein, and 30% to 40% fat—10% to 15% MUFA, <10% PUFA, and <10% SFA—from total daily energy intake; 20 to 35 g fiber per day; 50 to 75 GI). The InG and the CG followed calorie restriction (600 kcal/day) with the isocaloric match (1200 to 1500 kcal/day). No significant change in BW between the InG and the CG was found after the intervention. In contrast, the InG showed a significant increase in the ovulatory cycle compared with the CG [49] (Table 1).

In a randomized parallel pilot study by Atiomo et al., 2009 [50], obese women with PCOS who consumed a low-GI diet (InG; *n* = 6) for 6 months showed no differences in anthropometric parameters (BW, WC, HC, BMI, SBP, and DBP), glucose control (FBG and FBI), gonadal parameters (estradiol (ED), LH, SHBG, and T), lipids (TC, HDL-C, LDL-C, TG, and HDL-C/LDL-C ratio), and endometrial thickness compared with the control diet (CG; *n* = 5). The InG and the CG were isocaloric matched and calorie-restricted (600 kcal/day) [50] (Table 1).

In a 12-week RCT of overweight and/or obese women with PCOS by Mehrabani et al., 2012 [16], a low-GI plus high-protein diet (InG; *n* = 23; 40% carbohydrate, 30% protein, and 30% fat from total daily energy intake; <20 GL foods) significantly decreased WC, HC, FBI, and HOMA-IR compared with the control diet (CG; *n* = 26; 55% carbohydrate, 15% protein, and 30% fat from total daily energy intake). No significant differences were found in BW, percent of body fat (PBF), percent of LBM, suprailiac skinfold, adiponectin, gonadal parameters (total T, SHBG, A4, FSH, LH, dehydroepiandrosterone-sulfate (DHEAS), and FAI), lipids (TC, HDL-C, LDL-C, and TG), and inflammatory markers (high sensitivity C-reactive protein (hs-CRP), tumor necrosis factor-alpha (TNF- α), and interleukin-6 (IL-6)). The InG and CG were isocaloric matched and were instructed to restrict their calorie intake (500 to 1000 kcal/day) [16] (Table 1).

In summary of this review, three RCTs [16,49,50] investigated the effect of a calorie-restricted low-GI diet on PCOS in women with an isocaloric dietary design. The finding showed that a calorie-restricted low-GI diet was not associated with changes in BW [16,49,50], WC [16,50], HC [16,50], FBI [16,50], lipids (TC, LDL-C, HDL-C, and TG) [16,50], and gonadal parameters (total T, SHBG, and LH) [16,50]. Overall, the low-GI diet does not appear to have significant effects on body composition. However, the evidence for supporting the benefits of a low-GI diet for women with PCOS remains inconclusive, underscoring the need for further interventions to substantiate these findings.

#### 3.1.5. High-Protein Diet

Protein is an essential nutrient, playing a key role in the growth and development of the human body. Protein, an energy source, was recommended to be consumed in amounts of at least 0.80 g per kilogram of BW every day in adults to prevent deficiency. High-protein intake above this recommendation has been associated with loss of BW in clinical trials [75]. A high-protein diet is defined as providing more than 25% protein from total daily energy intake or 2.0 g protein/kg BW per day [76].

In this review, the five RCTs [13,14,15,51,52] examined the effect of a calorie-restricted high-protein diet on anthropometric parameters, glucose control, lipids, and gonadal parameters in overweight and/or obese women with PCOS. The five RCTs [13,14,15,51,52] included subjects aged 18 to 45 years, with a study duration ranging from 4 weeks to 16 weeks. Four [13,14,15,52] of five RCTs [13,14,15,51,52] investigated the effect of a calorie-restricted high-protein diet on PCOS in women with an isocaloric dietary design. One [51] of five RCTs [13,14,15,51,52] investigated the effect of a calorie-restricted high-protein diet on PCOS in women compared with a non-isocaloric-matched control diet.

In a randomized, open-labeled, placebo-controlled study by Dou et al., 2024 [51], overweight and/or obese women with PCOS were randomized to one of three groups for 8 weeks: calorie-restricted diet (CG; *n* = 22), calorie-restricted diet + whey protein (InG2; *n* = 30), calorie-restricted diet + whey protein + dietary fiber (InG1; *n* = 30). The CG (1000 to 1200 kcal/day) was advised to consume three meals (0.5 to 1 fist-sized portion of cooked staple foods (e.g., rice or noodles), 250 mL of low-fat milk, one egg, 100 g of leafy vegetables for breakfast, 100 g of lean meat with vegetables for lunch, and 100 g of bean products with vegetables for dinner) and two snacks (each snack: 100 g of fruit) per day. The InG1 consumed an additional 10 g of dietary fiber compared with the InG2. The InG1 and InG2 had lower BW, PBF, and MDA compared with the CG. The InG1 showed significantly decreased BMI and visceral fat area (VFA) compared with the InG2 and CG. Moreover, the InG1 and the InG2 significantly increased SOD compared with the CG. However, no significant differences were found in the FFM, FFM index, and HOMA-IR between the InG1, InG2, and CG [51] (Table 2).

Moran et al., 2010 [15] conducted a randomized parallel trial of overweight and/or obese women with PCOS. Women were randomized to either a high-protein diet (InG; *n* = 14; 43% carbohydrate, 27% protein, and 28% fat from total daily energy intake) or an isocaloric control diet (CG; *n* = 14; 57% carbohydrate, 16% protein, and 27% fat from total daily energy intake) for 16 weeks. The InG and the CG consumed a calorie-restricted diet (approximately 1400 kcal per day) for 12 weeks, followed by 4 weeks of a normal calorie diet. The InG and the CG showed decreased anthropometric parameters (BW, BMI, and mean arterial pressure), glucose control (FBI and insulin AUC), lipids (TG and free fatty acids (FFA)), and hs-CRP compared with the baseline [15]. However, no significant differences between the InG and the CG were found in the anthropometric parameters, glucose control, lipids, and hs-CRP (Table 1).

In a randomized parallel trial by Kasim-Karakas et al., 2009 [52], overweight and/or obese women with PCOS were randomly assigned to a calorie-restricted high-protein diet (InG; *n* = 11; 39.5% carbohydrate, 33.7% protein, and 26.2% fat from total daily energy intake) or an isocaloric-matched control diet (CG; *n* = 13; 56.7% carbohydrate, 16.6% protein, and 25.9% fat from total daily energy intake) for 2 months. The InG and the CG were advised to restrict calorie intake (450 kcal/day). The InG showed significant decreases in anthropometric parameters (BW, BMI, and TFM) and lipids (TC, HDL-C, and apoprotein B) compared with the CG, whereas no statistically significant differences were found in glucose control (FBG, FBI, HbA1c, HOMA-IR, leptin, and adiponectin), gonadal parameters (T, SHBG, FAI, and DHEAS), and TG [52] (Table 1).

In a randomized parallel trial by Stamets et al., 2004 [14], overweight and/or obese women with PCOS who consumed a high-protein diet (InG; *n* = 13; 40% carbohydrate, 30% protein, and 30% fat from total daily energy intake) for 4 weeks showed no significant reduction in anthropometric parameters (BW, HC, WHR, SBP, and DBP), glucose control (FBG/FBI ratio, glucose AUC, insulin AUC, fasting leptin, and leptin AUC), gonadal parameters (DHEAS, total T, bioavailable T, LH, and FSH), and lipids (TC, HDL-C, LDL-C, and TG) compared with an isocaloric control diet (CG; *n* = 13; 55% carbohydrate, 15% protein, and 30% fat from total daily energy intake). The InG and the CG diminished calorie intake (1000 kcal/day) and received multivitamins and mineral supplements [14]. (Table 1).

Moran et al., 2003 [13] conducted a randomized parallel study to investigate the effects of a high-protein diet on anthropometric parameters, glucose control, and lipids in overweight women with PCOS. The study randomly allocated to either a calorie-restricted high-protein diet (InG; *n* = 14; 40% carbohydrate, 30% protein, and 30% fat from total daily energy intake) or an isocaloric control diet (CG; *n* = 14; 55% carbohydrate, 15% protein, and 30% fat from total daily energy intake). The study period was 16 weeks, with a 12-week calorie restriction phase and a 4-week weight maintenance period. After 12 weeks of the calorie restriction phase, no significant differences in glucose control (FBG, FBI, and HOMA-IR) were found between the InG and the CG. The InG significantly decreased TC/HDL-C compared with the CG. There were no differences in TC, TG, and LDL-C between the InG and the CG. In addition, a calorie-restricted diet was not associated with a decrease in gonadal parameters (SHBG, total T, and FAI) between the InG and the CG. After the calorie restriction phase, subjects followed a 4-week weight maintenance diet with the same nutrient composition as a calorie-restricted diet. The InG decreased anthropometric parameters (BW, TFM, LBM, and abdominal fat mass) compared with the baseline, while no significant differences were found between the InG and the CG. Moreover, no differences in glucose control (FBG, FBI, HOMA-IR, glucose AUC, and insulin AUC) between the InG and the CG were observed. The effect of the InG on HDL-C and TC/HDL-C was unchanged after following a weight maintenance diet [13] (Table 1).

In summary of this review, five RCTs [13,14,15,51,52] investigated the effect of a high-protein diet on PCOS in overweight and/or obese women. The high-protein diet showed a significant decrease in BW compared with the control diet in two [51,52] of five RCTs [13,14,15,51,52]. Two [15,51] of three RCTs [15,51,52] found that a high-protein diet showed a significantly decreased BMI compared with the control diet. One [52] of two RCTs [13,52] found that a high-protein diet showed a significant decrease in TFM. In addition, no significant differences in FBG [13,15,52], FBI [13,52], HOMA-IR [13,51,52], and glucose AUC [14,15] were observed between the high-protein diet and the control diet. With regard to lipids, the high-protein diet was not associated with TG reduction compared with the control diet [13,14,15,52]. A significant decrease in TC was found in one [52] of three RCTs [13,14,52]. The high-protein diet showed no significant decrease in LDL-C in two RCTs [13,14]. The high-protein diet could have no beneficial role in the lipids of women with PCOS. There were no significant differences in total T [13,14,52], FAI [13,15,52], SHBG [13,52], and DHEAS [14,52] between the high-protein diet and the control diet. In conclusion, no significant differences in anthropometric parameters, glucose control, and gonadal parameters were observed between the high-protein diet and the control diet.

#### 3.1.6. Low-Calorie Diet

In this review, one RCT [53] examined the effect of a low-calorie diet on anthropometric parameters, glucose control, and gonadal parameters in women with PCOS.

In a randomized, open-labeled, placebo-controlled study by Deshmukh et al., 2023 [53], obese women with PCOS were randomized to either a very low-calorie diet (InG; *n* = 11) or a low-calorie diet (CG; *n* = 11) for 16 weeks (8 weeks of dietary intervention and 8 weeks of diet reintroduction). In the intervention period, the InG consumed a total of 800 kcal per day (each meal: 200 kcal, 21 g carbohydrate, 15 g protein, 3 to 4 g fat) with meal replacements, while the CG consumed a calorie-restricted diet (600 kcal deficit from estimated energy requirements). In the reintroduction period, the InG received a calorie-enhancing meal of which calorie increases in each 2-week period until it reaches 1600 kcal. The InG significantly decreased BW, BMI, WC, WHR, TFM, trunk fat, and FFM compared with the CG after the intervention period. A significant decrease in FBG was found in the InG compared with the CG, while no difference in 2 h glucose after oral glucose tolerance test (OGTT) was found. An increase in SHBG was observed in the InG compared with the CG, whereas no differences in total T, DHEAS, FAI, A4, LH, and FSH were observed [53] (Table 2).

### 3.2. Non-Calorie-Restricted Diet

This review included six RCTs [17,18,54,55,56,57] that investigated the effect of non-calorie-restricted dietary patterns on PCOS in women. The six RCTs [17,18,54,55,56,57] were an isocaloric-matched intervention design. The dietary patterns of the six RCTs [17,18,54,55,56,57] without calorie restriction include a low-carbohydrate diet [18,54,55,57], a low-GI diet [56], and a high-protein diet [17].

#### 3.2.1. Low-Carbohydrate Diet

In this review, four RCTs [18,54,55,57] investigated the effect of a non-calorie-restricted low-carbohydrate diet on anthropometric parameters, glucose control, lipids, and gonadal parameters with an isocaloric control diet in overweight and/or obese women with PCOS. The four RCTs [18,54,55,57] included subjects aged 19 to 50 years, with dietary interventions for a duration of 8 to 20 weeks.

Perelman et al., 2017 [57] conducted a randomized crossover study to assess if low intake of carbohydrates can ameliorate hyperinsulinemia in obese women with PCOS. Women with PCOS treated with a low-carbohydrate diet (InG; *n* = 6; 40% carbohydrate, 15% protein, and 45% fat from total daily energy intake) for 3 weeks showed no changes in BW, TC, HDL-C, and TG compared with a control diet (CG; *n* = 6; 60% carbohydrate, 15% protein, and 25% fat from total daily energy intake). LDL-C concentration was significantly lower in the InG compared with the CG. The InG showed a significant decrease in insulin AUC compared with CG, whereas no difference in postprandial glucose AUC response was observed [57] (Table 1).

Goss et al., 2014 [55] conducted a randomized crossover study to examine the effect of the low-carbohydrate diet on body composition compared with the isocaloric control diet. The overweight and/or obese women with PCOS were randomized to either a low-carbohydrate diet (InG; *n* = 27; 41% carbohydrate, 19% protein, and 40% fat from total daily energy intake; ~50 GI) or a control diet (CG; *n* = 23; 55% carbohydrate, 18% protein, and 27% fat from total daily energy intake; ~60 GI). The intervention phase of the InG and the CG lasted for a period of 8 weeks, followed by a washout period of 4 weeks. The InG showed a significant reduction in anthropometric parameters (BW, TFM, PBF, intra-abdominal adipose tissue, subcutaneous abdominal adipose tissue, thigh intramuscular adipose tissue, and thigh subcutaneous adipose tissue) compared with the baseline. However, no differences in anthropometric parameters were observed in the InG compared with the CG [55] (Table 1).

Gower et al., 2013 [18] conducted a randomized crossover study to examine the effect of a low-carbohydrate diet on beta-cell responsiveness and concentration of T and Si compared with the isocaloric control diet. The overweight and/or obese women with PCOS were randomly assigned to either a low-carbohydrate diet (InG; *n* = 27; 41% carbohydrate, 19% protein, and 40% fat from total daily energy intake; ~50 GI) or a control diet (CG; *n* = 23; 55% carbohydrate, 18% protein, and 27% fat from total daily energy intake; ~60 GI). The InG and the CG adhered to dietary intervention for a period of 8 weeks, followed by a washout period of 4 weeks. No significant difference in BW was observed between the InG and the CG. The InG exhibited significant reductions in basal beta-cell response to glucose (PhiB), HOMA-IR, FBG, and FBI and increases in dynamic beta-cell response to glucose (PhiD), the volume of insulin secreted in the first phase (X0), and Si compared with the baseline. Furthermore, significant reductions in total T, TC, and LDL-C were observed in the InG after the intervention, while TG and the ratio of TC/HDL-C showed no significant changes in the InG [18] (Table 1).

In a crossover study by Douglas et al., 2006 [54], overweight and/or obese women with PCOS were assigned to follow three isocaloric-matched diets (2000 to 2300 kcal/day) for 16 days each, followed by a 3-week washout period. The three dietary interventions are as follows: a low-carbohydrate diet (InG1; *n* = 11; 43% carbohydrate, 15% protein, and 45% total fat (8% SFA, 17% PUFA, and 18% MUFA) from total daily energy intake; 83 mg cholesterol and 29 g fiber per day), a MUFA diet (InG2; *n* = 11; 55% carbohydrate, 15% protein, and 33% fat (7% SFA, 6% PUFA, and 17% MUFA) from total daily energy intake; 108 mg cholesterol and 24 g fiber per day), and a control diet (CG; *n* = 11; 56% carbohydrate, 16% protein, and 31% fat (7% SFA, 10% PUFA, and 13% MUFA) from total daily energy intake; 115 mg cholesterol and 27 g fiber per day). The InG1 significantly decreased FBI and TC compared with the CG. In addition, the InG1 significantly decreased BW and acute insulin response to glucose (AIRg) compared with the InG2. The InG2 significantly increased A4 compared with the CG. No significant differences were observed across the groups in FBG, Si, disposition index, gonadal parameters (total T, free T, DHEAS, SHBG, LH, and FSH), and lipids (TG, FFA, HDL-C, and LDL-C) [54] (Table 1).

In summary of this review, four RCTs [18,54,55,57] examined the effect of a non-calorie-restricted low-carbohydrate diet on PCOS in women with an isocaloric dietary design. The low-carbohydrate diet significantly decreased BW compared with the control diet in one [54] of four RCTs [18,54,55,57]. With regard to glucose control, significant reductions in FBI and AIRg were found in the low-carbohydrate diet compared with the control diet, while FBG and Si were not significant [54]. The low-carbohydrate diet significantly reduced insulin AUC compared with the control diet, while no significant difference in glucose AUC was found [57]. The low-carbohydrate diet decreased TC compared with the control diet in one [57] of two RCTs [54,57]. Moreover, the low-carbohydrate diet decreased LDL-C compared with the control diet in one [54] of two RCTs [54,57]. There were no significant differences in HDL-C and TG between the low-carbohydrate diet and the control diet [54,57]. One RCT [54] showed no significant changes in total T, free T, SHBG, DHEAS, LH, or FSH. The low-carbohydrate diet was found to have no significant association with BW reduction, glucose control, and lipids in women with PCOS.

#### 3.2.2. Low-GI Diet

In this review, one RCT [56] investigated the effect of a non-calorie-restricted low-GI diet on weight loss and glucose control with an isocaloric control diet in women with PCOS.

In a randomized crossover study by Hoover et al., 2021 [56], overweight and/or obese women with PCOS were randomly allocated to one of two dietary patterns: a non-calorie-restricted low-GI diet (InG; *n* = 27; 41% carbohydrate, 19% protein, and 40% fat from total daily energy intake; ~50 GI) or an isocaloric control diet (CG; *n* = 27; 55% carbohydrate, 18% protein, and 27% fat from total daily energy intake; ~60 GI). Both the InG and the CG adhered to the 8 weeks of each dietary intervention, followed by a 4-week washout period. The InG and the CG showed significant reductions in FBG, FBI, ghrelin, and cortisol and an increase in glucagon-like peptide-1 (GLP-1) compared with the baseline. However, no significant differences in BMI, FBG, FBI, glucagon, GLP-1, ghrelin, cortisol, and peptide YY were observed between the InG and the CG [56] (Table 1).

#### 3.2.3. High-Protein Diet

This review included only one RCT [17] that examined the effect of a non-calorie-restricted high-protein diet on PCOS in women with an isocaloric dietary design.

In a randomized parallel trial by Sørensen et al., 2012 [17], women with PCOS were assigned to one of two dietary patterns, a non-calorie-restricted high-protein diet (InG; *n* = 14; <30% carbohydrate, >40% protein, and 30% fat from total daily energy intake) and an isocaloric control diet (CG; *n* = 13; >55% carbohydrate, <15% protein, and 30% fat from total daily energy intake), for 6 months. The InG and the CG were instructed to maintain their respective nutrient composition without calorie restriction. The findings showed that the InG significantly reduced BW, BMI, TFM, WC, and FBG compared with the CG. However, no significant differences were found in C-peptide levels, gonadal parameters (total T, free T, and SHBG), and lipids (TC, HDL-C, LDL-C, TG, TC/HDL-C ratio, and TG/HDL-C ratio) between the InG and the CG [17] (Table 1).

## 4. Discussion

This review determined which dietary pattern is most effective for PCOS in women, separately investigating RCTs with an isocaloric or a non-isocaloric dietary design. This review highlights 21 RCTs [13,14,15,16,17,18,19,20,23,24,47,48,49,50,51,52,53,54,55,56,57] that examined the effects of dietary patterns on PCOS in women. There were 18 RCTs [13,14,15,16,17,18,19,20,23,24,48,49,50,52,54,55,56,57] with isocaloric dietary design and 3 RCTs [47,51,53] with non-isocaloric dietary design. There were 15 RCTs with a calorie-restricted dietary design and 6 RCTs with a non-calorie-restricted dietary design. The findings indicate that a calorie-restricted DASH diet had a significant effect on weight loss and glucose control in women with PCOS. A low-carbohydrate diet, low-GI diet, and low-protein diet showed neutral effects on anthropometric parameters, glucose control, lipids, and gonadal parameters in women with PCOS.

### 4.1. Calorie-Restricted DASH Diet

In this review, four RCTs [19,20,23,24] examined the effect of a calorie-restricted DASH diet on PCOS in women with an isocaloric-matched control diet. The DASH diet and the control diet had the same nutrient composition, ranging from 50 to 55% carbohydrate, 15 to 20% protein, and 25 to 30% fat from total daily energy intake for an intervention period. In addition, the total daily energy intake was restricted by the distribution of BMI, ranging from 350 to 700 kcal per day. The findings showed that a calorie-restricted DASH diet significantly decreased BW and BMI in overweight and/or obese women with PCOS compared with the control diet [19,20,23,24]. Moreover, a calorie-restricted DASH diet significantly decreased FBI [20,24] and HOMA-IR [20,24], with an increase in QUICKI [20,24]. However, no significant difference was found in FBG between the calorie-restricted DASH diet and the control diet in two RCTs [20,24]. No significant difference in lipids was found between the calorie-restricted DASH diet and the control diet [19]. Moreover, the calorie-restricted DASH diet showed no difference in total T compared with the control, while a significant increase in SHBG was found [23,24].

In line with these findings, the DASH diet [19,20,23,24] was found to be favorable in managing glucose control (HOMA-IR, FBG, and FBI) in a network meta-analysis of 19 RCTs by Juhász et al., 2024 [77]. In addition, they also found that the DASH diet was an effective dietary pattern in TG reduction [77]. Although the DASH diet showed high importance in glucose control and TG reduction, no significant differences in FBI, FBG, and TG were found between the DASH diet and the control diet [77]. An umbrella review of meta-analyses of RCTs by Moslehi et al., 2023 [78] showed that a calorie-restricted DASH diet decreased BW (mean difference (MD) = −1.87 kg; 95% CI −2.64 to −1.11; *I*^2^ = 58%; low certainty), BMI (MD = −0.72 kg/m^2^; 95% CI −1.03 to −0.42; *I*^2^ = 62.6%; low certainty), FBI (MD = −4.67 μIU/mL; 95% CI −5.51 to −3.83; *I*^2^ = 73.8%; low certainty), and HOMA-IR (MD = −1.20; 95% CI −1.48 to −0.92; *I*^2^ = 12.5%; low certainty) in women with PCOS.

There is some evidence that the DASH diet improves the metabolic characteristics of women with PCOS [19,20,23,24]. Clinical trials on women with PCOS showed that weight loss from the DASH diet enhanced Si [20,24]. The weight loss reduced leptin [79] and increased concentrations of insulin-like growth factor-binding protein-1 [80] and adiponectin [81]. The excess adiponectin level stimulated the downregulation of adipose tissue [81]. In addition, the dietary antioxidants, mainly from fruits and vegetables rich in the DASH diet, alleviated inflammation and oxidative stress [62,82].

Therefore, the beneficial effects of a DASH diet on weight loss and glucose control were found in women with PCOS. Further interventions with larger sample sizes and longer intervention periods are needed to reveal the role of the DASH diet on PCOS in women.

### 4.2. Calorie-Restricted Ketogenic Diet

This review found one RCT [47] that examined the effect of a calorie-restricted KD on anthropometric parameters, glucose control, lipids, and gonadal parameters in overweight and/or obese women with PCOS compared with the calorie-restricted MED diet. The findings showed that the calorie-restricted KD had beneficial effects on anthropometric parameters (BW, BMI, WC, HC, WHR, TFM, FFM, basal metabolism, and total body water), glucose control (FBG, FBI, HOMA-IR, and C-peptide), lipids (TG, TC, and LDL-C), and gonadal parameters (total T, free T, LH, FSH, and SHBG) compared with the calorie-restricted MED diet [47].

Consistent with this review, a meta-analysis of 11 human interventions by Xing et al., 2024 [83] found a beneficial effect of the KD on weight loss in overweight or obese women with PCOS. The adherence to the KD showed a significant decrease in BW (weighted mean difference (WMD) = −9.13 kg; 95% CI −11.88 to −6.39, *p* < 0.001, *I*^2^ = 87.23%) [83]. In addition, the KD was associated with higher decreases in BMI (WMD = −2.93 kg/m^2^; 95% CI −3.65 to −2.21; *p* < 0.01; *I*^2^ = 78.81%), WC (WMD = −7.62 cm; 95% CI −10.73 to −4.50; *p* < 0.01; *I*^2^ = 89.17%), and TFM (WMD = −6.62 kg; 95% CI −8.44 to −4.80; *p* < 0.01; *I*^2^ = 53.92%) compared with the control diet [83]. This meta-analysis [83] included only three interventions [47,84,85] with an RCT design and eight interventions [12,86,87,88,89,90,91,92] with a pre-post design.

Although the KD has been demonstrated to have a favorable impact on anthropometric parameters, glucose control, and gonadal parameters in women with PCOS, the mechanisms underlying its beneficial effects remain to be elucidated. The KD stimulates nutritional ketosis through the restriction of carbohydrates, thereby increasing lipolysis and enhancing metabolic efficiency [93]. It has been postulated that the restriction of carbohydrates improves Si, thereby enhancing endocrine function [93]. Moreover, the insulin level has been associated with androgen levels and SHBG synthesis in women with PCOS [94].

In summary of this review, the KD was found to have a beneficial role in the management of anthropometric parameters, glucose control, lipids, and gonadal parameters in overweight and/or obese women with PCOS compared with the control diet. Although the RCT recruited had a large number of subjects (*n* = 144), the short intervention period can be regarded as a limitation. Furthermore, the number of relevant RCTs is relatively small, highlighting the necessity of future interventions with an isocaloric dietary design to draw a conclusion.

### 4.3. Low-Carbohydrate Diet

This review examined the effect of a low-carbohydrate diet on weight loss, glucose control, lipids, and gonadal parameters in women with PCOS from five RCTs [18,48,54,55,57]. One [48] of five RCTs [18,48,54,55,57] conducted a dietary intervention with calorie restriction. Women with PCOS consumed the same calories between the low-carbohydrate diet and the control diet in the five RCTs [18,48,54,55,57]. The consumption of carbohydrates accounted for approximately 40% of the total daily energy intake in four RCTs [18,54,55,57] with a non-calorie-restricted dietary design.

This review found that a low-carbohydrate diet had a neutral effect on weight loss in four [18,48,55,57] of five RCTs [18,48,54,55,57]. Consistent with this review, Hashimoto et al., 2016 [95] showed that a low-carbohydrate diet (about 40% of total daily energy intake) was not associated with BW [standardized mean difference (SMD) = −0.37 kg; 95% CI −0.85 to 0.12] and TFM (SMD = −0.65 kg; 95% CI −1.32 to 0.02) reductions in overweight or obese subjects. In contrast, the strict limitation of carbohydrates (about 10% of total daily energy intake) significantly decreased BW (SMD = −1.00 kg; 95% CI −1.54 to −0.45) and TFM (SMD = −0.97 kg; 95% CI −1.50 to −0.44) compared with the control diet [95]. Similarly, Mansoor et al., 2016 [96] showed that a low carbohydrate diet (<20% of total daily energy intake) significantly decreased BW (WMD = −2.17 kg; 95% CI −3.36 to −0.99) compared with the low-fat diet. Furthermore, a low-carbohydrate diet (<45% of total daily energy intake) decreased BMI (MD = −0.99 kg/m^2^; 95% CI −1.83 to −0.15; *I*^2^ = 0%; low certainty) compared with the control diet [78].

The beneficial effect of a low-carbohydrate diet on glucose control was inconsistent in women with PCOS in this review. Moslehi et al., 2023 [78] found that a low-carbohydrate diet significantly decreased FBI (MD = −3.40 μIU/mL; 95% CI −6.03 to −0.77; *I*^2^ = 73.8%; very low certainty) compared with the control diet. In addition, a meta-analysis by Porchia et al., 2020 [97] found that a low-carbohydrate diet (<50% of total daily energy intake) significantly reduced insulin resistance (standard paired differences = −0.587; 95% CI −0.812 to −0.362; *p* < 0.001) in women with PCOS.

In this review, the low-carbohydrate diet significantly decreased TC [54] and LDL-C [57] in two RCTs [54,57] compared with an isocaloric control diet, while no significant differences in HDL-C and TG were observed [54,57]. Juhász et al., 2024 [77] found that the low-carbohydrate diet showed the most favorable effect on cholesterol reduction in women with PCOS. However, no significant difference in cholesterol reduction was found between a low-carbohydrate diet and a normal diet [77]. A significant difference in TC (MD = −8.37 mg/dL; 95% CI −16.6 to −0.14; *I*^2^ = 20.5%; low certainty) was observed between the low-carbohydrate diet and the control diet in an umbrella review of meta-analyses by Moslehi et al., 2023 [78].

With regard to gonadal parameters, a low-carbohydrate diet showed no significant differences in total T [48,54], free T [48,54], SHBG [48,54], DHEAS [54], FAI [48], LH [54], and FSH [54] compared with the control diet in women with PCOS.

A low-carbohydrate diet has been shown to bring beneficial effects on weight loss and ovarian function, reducing concentrations on FBG, FBI, insulin-like growth factor-1, and insulin-like growth factor-binding protein-1 [98]. In addition, a low-carbohydrate diet was found to decrease TG and increase HDL-C compared with the high-carbohydrate diet [99]. Moreover, a low-carbohydrate diet restores the balance of inositol metabolism, which can increase Si [100]. This can help improve insulin resistance effectively, lower androgen levels, and enhance the menstrual cycle in women with PCOS [101].

In summary of this review, a low-carbohydrate diet had neutral effects on weight loss, glucose control, and lipids in women with PCOS. The favorable role of a low-carbohydrate diet on PCOS was inconsistent across the restriction of carbohydrate intake, highlighting that further interventions with carbohydrate consumption of less than 40% of total daily energy intake are needed. Four [18,54,55,57] of five RCTs [18,48,54,55,57] indicated a small sample size as the limitation. In addition, the limitation of two RCTs [54,57] was a short-term intervention period. Further interventions with large sample sizes and longer intervention periods are needed to clarify the evidence in women with PCOS.

In a randomized parallel trial by Mei et al., 2022 [102], overweight women with PCOS who consumed a calorie-restricted dietary pattern consisting of a MED diet plus a low-carbohydrate diet (<20% carbohydrate, up to 100 g carbohydrate) for 12 weeks showed significant decreases in anthropometric parameters, glucose control (FBG, FBI, HOMA-IR, and QUICKI), lipids (TG, TC, and LDL-C), and gonadal parameters (LH and LH/FSH), compared with the control low-fat diet (<40 g fat (<30% from total daily energy intake), up to 10% SFA). PCOS women (*n* = 30) who followed the intervention dietary pattern were advised to increase the intakes of protein and fat, whole grain (high in extra virgin olive oil, green leafy vegetables, fruit, cereals, nuts, and legumes; moderate in fish, other meat, dairy products; low in eggs), while PCOS women (*n* = 29) who followed the control diet were advised to increase intake of cereals, vegetables, and fruits and avoid fatty foods (fatty meats, butter, offal, fried foods, preserved foods, poultry skin, fish roe, shrimp roe, and crab meat). The intervention diet and control diet were isocaloric matched [102].

### 4.4. Low-GI Diet

In this review, three RCTs [49,50,56] examined the effect of a low-GI diet on anthropometric parameters, glucose control, lipids, and gonadal parameters in women with PCOS aged from 18 to 51 years. Two [49,50] of three RCTs [49,50,56] restricted the calorie intake of the women during the intervention. The GI cutoffs for the low-GI diet in three RCTs [49,50,56] were as follows: one RCT [56] had a GI cutoff of approximately 50, another RCT [49] had a maximum of 45 GI, while the third RCT [50] did not specify any GI threshold. This review found that the low-GI diet does not play a vital role in the management of anthropometric parameters, glucose control, and lipids in women with PCOS.

In contrast with this review, an umbrella review of meta-analyses of RCTs by Moslehi et al., 2022 [78] found the beneficial effects of a low-GI diet on anthropometrics, glucose control, lipids, and gonadal parameters compared with the control diet in women with PCOS. A low-GI/GL diet decreased BW (MD = −1.51 kg; 95% CI −2.17 to −0.85; *I*^2^ = 8.6%; low certainty), WC (MD = −2.88 cm; 95% CI −5.21 to −0.55; *I*^2^ = 63.6%; very low certainty), FBI (MD = −2.93 μIU/mL; 95% CI −4.82 to −1.03; *I*^2^ = 71.0%; low certainty), HOMA-IR (MD = −0.76; 95% CI −1.29 to −0.23; *I*^2^ = 76.3%; low certainty), TC (MD = −8.81 mg/dL; 95% CI −16.0 to −1.60; *I*^2^ = 15.2%; low certainty), and low-density lipoprotein cholesterol (LDL-C) (MD = −8.98 mg/dL; 95% CI −15.5 to −2.48; *I*^2^ = 6.1%; low certainty). In particular, the consumption of a lower GI/GL diet for 8 to 24 weeks significantly decreased total T (MD = −0.07 ng/mL; 95% CI −0.10 to −0.04; *I*^2^ = 0%; low certainty) [78].

A meta-analysis of 10 RCTs [18,19,20,23,24,50,55,103,104,105] by Kazemi et al., 2021 [106] found the beneficial effects of a lower-GI diet on glucose control and lipids compared with the higher-GI diet. The lower-GI diet attenuated HOMA-IR compared with the higher-GI diet (WMD = −0.78; 95% CI −1.20 to −0.37; *p* < 0.001) [18,20,24,105] with high heterogeneity (*I*^2^ = 86.6%; *p* < 0.001). There were no differences in FBG and FBI between the lower-GI diet and the higher-GI diet [18,20,24,50,105] with high heterogeneity. The lower-GI diet attenuated TC (WMD = −11.13; 95% CI −18.23 to −4.04; *p* = 0.002), LDL-C (WMD = −6.27; 95% CI −12.01 to –0.53; *p* = 0.03), and TG (WMD = −14.85; 95% CI −28.75 to −0.95; *p* = 0.03) compared with the higher-GI diet, while no change in HDL-C was observed between the lower-GI diet and the higher-GI diet [18,19,50,105]. No effects on BW [19,23,24,50,55,104,105] and FAI [18,23,24,103] were found in the comparison between the lower-GI diet and the higher-GI diet. However, the lower-GI diet attenuated WC compared with the higher-GI diet (WMD = −2.81; 95% CI −4.40 to −1.23) [20,23,50,105] with moderate heterogeneity (*I*^2^ = 53.9%; *p* = 0.08). The lower-GI diet attenuated total T compared with the higher-GI diet (WMD = −0.21; 95% CI −0.32 to −0.09; *p* < 0.001) [18,23,24,50,103] with a low heterogeneity (*I*^2^ = 8.6%; *p* = 0.36) [106]. However, this meta-analysis [106] does not solely focus on the effect of a lower-GI diet on PCOS in women, including three RCTs with a DASH diet [20,23,24] and two RCTs with a low-carbohydrate diet [18,55], making it difficult to determine the clear effect of a low-GI diet on PCOS in women.

Saadati et al., 2021 [107] performed a meta-analysis of eight RCTs with study periods ranging from 3 to 12 months to investigate the effects of the low-GI diet on PCOS in women. The low-GI diet reduced 2 h insulin (four RCTs, 177 subjects, SMD = −0.79; 95% CI −1.33 to −0.24; *p* = 0.005; *I*^2^ = 57%) and HOMA-IR (six RCTs, 293 subjects, SMD = −0.36; 95% CI −0.59 to −0.12; *p* = 0.003; *I*^2^ = 0%) compared with the control diet. However, no difference in FBG (7 RCTs, 280 subjects), FBI (five RCTs, 192 subjects), 2 h glucose (four RCTs, 177 subjects), HbA1c (two RCTs, 114 subjects), and updated homeostasis model assessment of insulin sensitivity (HOMA2-IS) (two RCTs, 138 subjects) between the low-GI diet and the control diet. The low-GI diet reduced TG (six RCTs, 337 subjects, SMD = −0.02; 95% CI −0.50 to −0.07; *p* = 0.009; *I*^2^ = 0%), TC (six RCTs, 337 subjects, SMD = −0.48; 95% CI −1.07 to −0.11; *p* = 0.04; *I*^2^ = 68%), and HDL-C (five RCTs, 307 subjects, SMD = −0.92; 95% CI −1.31 to −0.53; *p* < 0.0001; *I*^2^ = 59%) compared with the control diet. The low-GI diet attenuated LH (four RCTs, 215 subjects, SMD = −0.30; 95% CI −0.57 to −0.03; *p* = 0.03; *I*^2^ = 27%) and T (six RCTs, 329 subjects, SMD = −0.52; 95% CI −0.83 to −0.22; *p* = 0.0009; *I*^2^ = 42%) compared with the control diet, while there were no changes in DHEAS (four RCTs, 193 subjects) and prolactin (three RCTs, 165 subjects) between the low-GI diets and the control diet. The low-GI diet reduced WC (two RCTs, 110 subjects, SMD = −6.16; 95% CI −10.12 to −2.20; *p* = 0.002; *I*^2^ = 0%) and TFM (two RCTs, 110 subjects, SMD = −2.90; 95% CI −3.63 to −2.17; *p* < 0.0001; *I*^2^ = 0%) compared with the control diet. However, no difference was found between the two diets for BMI (eight RCTs, 258 subjects), BW (eight RCTs, 298 subjects), PBF (three RCTs, 129 subjects), SBP (two RCTs, 80 subjects), DBP (two RCTs, 80 subjects), and LBM (two RCTs, 110 subjects). Fertility was higher in the low-GI diet compared with the control diet (three RCTs, 132 subjects, SMD = 1.45; 95% CI 0.30 to 2.61; *p* = 0.01; *I*^2^ = 79%). However, this meta-analysis [107] included one RCT [19] that investigated dietary interventions of a DASH diet.

Previous studies have hypothesized that a low-GI diet could play a significant role in glucose homeostasis, thereby alleviated the anthropometric and metabolic profile of women with PCOS [108]. The high-GI food-induced insulin elevation increased the lipogenesis in the adipose tissue [73]. A low-GI diet led to a gradual increase in postprandial glycemia, lowering insulin resistance [74] and the risk of PCOS exacerbation [109]. Short-term use of a low-GI diet could slightly improve Si in women with PCOS [110].

The number of RCTs with the low-GI diet intervention was insufficient to investigate the association between the low-GI diet and PCOS. More future interventions with a low-GI diet are needed to observe the evidence of the beneficial role of a low-GI diet.

### 4.5. High-Protein Diet

This review observed the findings from six RCTs [13,14,15,17,51,52] that examined the effect of a high-protein diet on anthropometric parameters, glucose control, lipids, and gonadal parameters in women with PCOS aged 18 to 45 years. In four RCTs [13,14,15,52], the contribution of protein intake to total daily energy intake ranged from approximately 27% to 33.7%. The contribution of protein intake exceeded 40% of total daily energy intake in one RCT [17]. In another RCT [51], the high-protein diet group consumed a protein of 1.5 to 2 g/kg per day. Five [13,14,15,51,52] of six RCTs [13,14,15,17,51,52] restricted the calorie intake of the subjects during dietary intervention. Five [13,14,15,17,52] of six RCTs [13,14,15,17,51,52] advised consumption of the same calories in the high-protein diet and the control diet. Five RCTs [13,14,15,17,52] with an isocaloric-matched design showed a neutral effect of a high-protein diet on BW change compared with the control diet. Two [17,52] of three RCTs [15,17,52] with an isocaloric-matched design showed a significant reduction in BMI compared with the control diet. One RCT [51] with a non-isocaloric matched design showed a beneficial effect of a high-protein diet on BW and BMI reductions compared with the control diet.

A high-protein diet has been suggested as effective treatment for weight loss [111]. In addition, a high-protein diet was associated with the reductions in HOMA-IR, LDL-C, TC, and TG in individuals with T2DM [112]. Furthermore, a high-protein diet significantly decreased FBI and HOMA-IR in women with PCOS [111].

In summary, this review found that no significant differences in anthropometric parameters (BW, LBM, and TFM), glucose control (FBG, FBI, and HOMA-IR), lipids (TC, HDL-C, LDL-C, TG, TC/HDL-C ratio, and TG/HDL-C ratio), and gonadal parameters (total T, free T, SHBG, DHEAS, and FAI) between the high-protein diet and the control diet were observed in both an isocaloric and non-isocaloric design. Although the intervention periods of RCTs in this review were sufficient (4 weeks to 6 months), the sample size was small (*n* < 30) in five [13,14,15,17,52] of six RCTs [13,14,15,17,51,52].

### 4.6. Low-Calorie Diet

This review included one RCT [53] that examined the effect of a very low-calorie diet on anthropometric parameters, glucose control, lipids, and gonadal parameters in overweight and/or obese women with PCOS compared with the low-calorie diet. The findings showed that the very low-calorie diet significantly decreased anthropometric parameters (BW, BMI, WC, WHR, TFM, trunk fat, and FFM) and FBG compared with the low-calorie diet [53]. The number of RCTs with the low-calorie diet intervention was not sufficient, as well as the sample size, to examine the association between the low-calorie diet and PCOS in women.

In a network meta-analysis by Juhász et al., 2024 [77], the low-calorie diet was the most effective dietary pattern in BMI reduction. Additionally, they found that a low-calorie diet with metformin showed the highest efficiency in weight loss [77]. Yang et al., 2024 [113] investigated the impact of dietary patterns on PCOS women with BMI ≥ 25 kg/m^2^ by conducting a meta-analysis of RCTs that addressed a calorie-restricted diet, a low-calorie combined with a low-carbohydrate diet, and a low-calorie diet with extracts. The meta-analysis of RCTs found that BW [25,53,114,115,116,117], BMI [25,53,114,115,116,117,118,119], WC [53,115,116,117], HC [116,117], and WHR [53,115,116] were improved after the dietary intervention compared with the control. Levels of FBG [53,115,118,119,120], FBI [115,118,119,120], HOMA-IR [115,117,118,119,120], LDL-C [117,118], TC [53,117,118], LH [25,53,114,115,117,118,119], and ED [117,118,119] were improved after the dietary intervention compared with control.

### 4.7. Other Interventions in Women with PCOS and Without PCOS

This review focused on RCTs that included women with PCOS as participants to examine the effect of dietary patterns on PCOS in women. Therefore, the five interventions [121,122,123,124,125] that included women without PCOS were excluded from this review. However, overweight or obese women without PCOS share the characteristics of women with PCOS, which had a higher risk of complications such as T2DM, obesity, and cardiovascular disease.

Of the five interventions [121,122,123,124,125], three interventions [121,122,123] investigated the impact of dietary patterns on PCOS in women compared with non-PCOS women. In comparison, two RCTs [124,125] investigated the effect of dietary patterns on anthropometric parameters, glucose control, lipids, and gonadal parameters in both women with and without PCOS.

Shishehgar et al., 2023 [121] conducted an intervention study to assess the impact of a calorie-restricted diet on anthropometric parameters and quality of life in overweight and/or obese women with PCOS and overweight and/or obese women without PCOS. The subjects included 105 PCOS women (PCOS group (PG)) and 111 non-PCOS women (non-PCOS group (NPG)), who followed a calorie-restricted low-GI diet (500 kcal deficit; 50% low-GI carbohydrate, 20% protein, and 30% fat). After a 24-week intervention period, significant decreases in BW, BMI, WC, and WHR were observed in the PG and the NPG compared with the baseline. The PG showed a higher reduction in WC than the NPG. However, no significant differences in BW, BMI, and WHR between the PG and the NPG were observed. Additionally, the PG and the NPG showed an improvement in health-related quality of life in comparison to their baseline scores [121].

In an intervention study by Shishehgar et al., 2019 [122], 28 overweight and/or obese women with PCOS (PG) and 34 overweight and/or obese women without PCOS (NPG) were instructed to follow an isocaloric-matched low-GI diet (50% low and medium GI carbohydrate, 20% protein, and 30% fat) with calorie restriction (500 kcal deficit). After 24 weeks of intervention, there were no significant differences in BW, BMI, WC, WHR, and SBP between the PG and the NPG. The PG had a higher decrease in SBP compared with the NPG. The PG and the NPG showed significant decreases in FBI and HOMA-IR compared with the baseline, while no differences between the PG and the NPG were found. No significant difference was observed in FBG. The PG showed a significant increase in SHBG and decreases in total T and FAI compared with the baseline. In addition, menstrual cycle, frequency, and irregularities were significantly improved in the PG compared with the baseline [122].

An RCT by Moran et al., 2007 [123] investigated the effects of a calorie-restricted diet on BW and inflammatory biomarkers in overweight women with PCOS and overweight women without PCOS. A total of 32 women with PCOS and without PCOS followed a calorie-restricted diet (~1440 kcal; 57.2% carbohydrate, 20.1% protein, and 22.7% fat from total daily energy intake) for 8 weeks. Subjects were instructed to consume two meals with replacements and one meal with a low-fat meal per day. There were no significant differences in anthropometric parameters (BW, WC, TFM, and FFM), glucose control (adiponectin, FBI, and HOMA-IR), lipids (TC, LDL-C, HDL-C, and TG), gonadal parameters (total T, SHBG, FAI, and free T), and inflammation markers (IL-6, TNF-α, and CRP) between women with PCOS and women without PCOS [123].

In a single-blind randomized controlled study by Toscani et al., 2011 [124], 22 women with PCOS and 18 women without PCOS were randomly assigned to follow a calorie-restricted high-protein diet (InG; 40% carbohydrate, 30% protein, and 30% fat from total daily energy intake) or an isocaloric control diet (CG; 55% carbohydrate, 15% protein, and 30% fat from total daily energy intake). The daily calorie consumption was determined by the BW (normal BW: 25 to 30 kcal/kg of current BW; overweight or obese: 20 to 25 kcal/kg of current BW). After the 8-week intervention, both InG and CG decreased BW, BMI, WC, PBF, and trunk skinfolds compared with the baseline. However, no differences in BMI, WC, PBF, and trunk skinfolds between the InG and the CG were found. There were no changes in FBG, FBI, HOMA-IR, glucose AUC, and insulin AUC in the InG and the CG. In addition, no significant differences in TC, LDL-C, HDL-C, non-HDL-C, and TG were observed between the InG and the CG. The total T significantly showed a significant decrease in the InG and the CG compared with the baseline, while the differences in SHBG and FAI were not significant. There were no significant differences in anthropometric parameters, glucose control, lipids, and gonadal parameters between women with PCOS and women without PCOS [124].

In an RCT by Moran et al., 2004 [125], 20 overweight women with PCOS and 12 overweight women without PCOS were randomly advised to follow either a high-protein diet (InG; 40% carbohydrate, 30% protein, and 30% fat from total daily energy intake) or an isocaloric-matched control diet (CG; 55% carbohydrate, 15% protein, and 30% fat from total daily energy intake) for 12 weeks. The InG and the CG were instructed to restrict calorie intake for 12 weeks (~14,340 kcal/day), followed by a 4-week weight maintenance diet with the same macronutrient composition. The differences in BW, FBG, FBI, HOMA-IR, fasting leptin, and insulin AUC were not significant between the InG and the CG, whereas the CG had a higher glucose AUC compared with the InG at baseline and 16-week. No significant differences were found between women with PCOS and those without PCOS. There were no significant differences in fasting ghrelin and ghrelin response between the InG and the CG. However, women without PCOS had a higher increase in fasting ghrelin and a decrease in ghrelin response compared with those with PCOS [125].

### 4.8. Strengths and Limitations

This review has several advantages. This review had the strict inclusion criteria for RCTs with the following characteristics: women with PCOS, adults aged 18 years and over, intervention with dietary patterns, excluding single foods, extracts, supplements, or medications, and non-pre-post interventions. This review further excluded RCTs that included subjects who had a history of metformin use to control the confounding effect of medications. These could reduce bias and focus on the effect of dietary patterns on PCOS in women. This review was to figure out the beneficial effect of dietary patterns on PCOS, categorizing the outcome of RCTs with calorie restriction or isocaloric matching.

Nonetheless, this review has limitations. No risk of bias methods were used to assess the included RCTs. There are limited numbers of included RCTs that investigate the effect of dietary patterns on PCOS in women. In addition, the small number of subjects of RCTs makes it difficult to generalize the findings from this review. The majority of the RCTs recruited women with PCOS in Western countries (USA (*n* = 7), Iran (*n* = 5), Australia (*n* = 3), UK (*n* = 2), Italy (*n* = 1), Denmark (*n* = 1), Mexico (*n* = 1), China (*n* = 1), Africa (*n* = 0), and Latin America (*n* = 0)), limiting the generalization of the findings from this review to other regions. In 17 of 21 RCTs included in this review, overweight and/or obese women with PCOS were recruited, which indicates that the findings may not represent PCOS women with normal BW. Future interventions that include normal-weight women with PCOS are needed.

## 5. Conclusions

In conclusion, a calorie-restricted DASH diet appeared to have beneficial roles in weight loss and glucose control in women with PCOS. The beneficial roles of a low-carbohydrate diet, a low-GI diet, and a high-protein diet were unclear based on the inconsistent findings. The KD and a low-calorie diet seem to have a vital role in PCOS, while the evidence is limited. These findings suggest that dietary composition could play an important role in the management of PCOS in women. Especially, adherence to the DASH dietary pattern has been suggested as a potential treatment approach for weight loss and glucose control. However, in light of studies [126,127,128], ethnic or regional differences could play an important role in the process of dietary patterns affecting the prognosis of PCOS. Further intervention studies differing the ethnic or regional backgrounds of the subjects from different countries are necessary to verify the role of dietary patterns on the prognosis of PCOS in women, which will enable the establishment of subsequent dietary policy recommendations for PCOS women of different ethnic and regional backgrounds.

## Figures and Tables

**Figure 1 nutrients-17-00674-f001:**
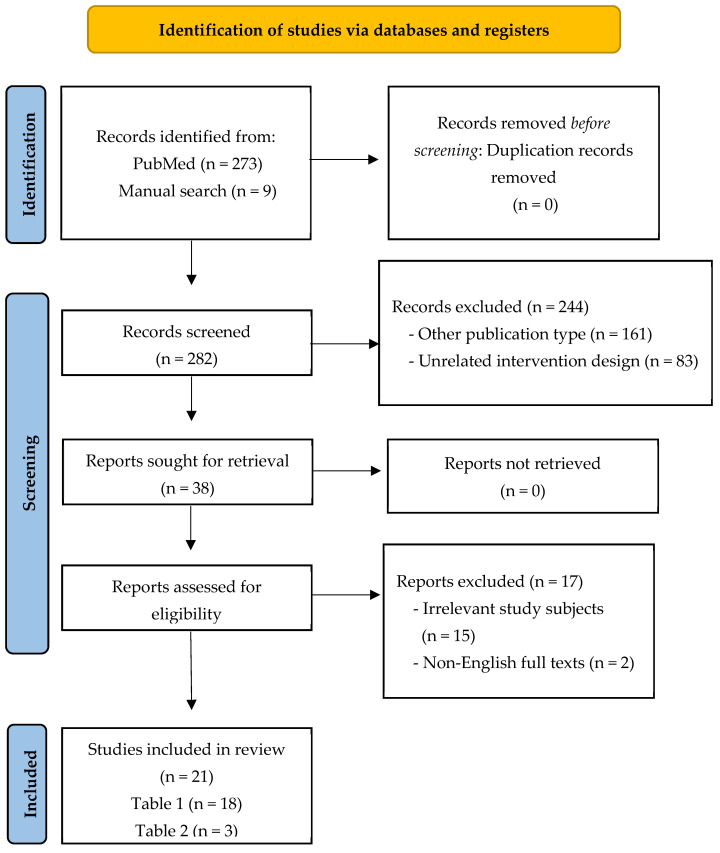
The flow diagram for the screening and selection process of this review.

**Table 1 nutrients-17-00674-t001:** Effects of dietary patterns in individuals with polycystic ovary syndrome in the isocaloric dietary RCTs.

Dietary Pattern	Reference	Region	Study Design	Subject Health Status	Dietary Intervention	Mean Age (Years)	Study Period	Outcomes
DASH diet	Azadi-Yazdi et al., 2017 [23]	Iran	Parallel	Overweight and/or obese women with PCOS	InG (*n* = 28): DASH diet (50 to 55% carbohydrate, 15 to 20% protein, and 25 to 30% fat from total E; sodium < 2400 mg/day) Dietary components (1600 kcal/day): grains: 6 servings (at least 3 servings from whole grains), simple sugar: 2 servings, vegetables: 4 servings, fruits: 4 servings, dairy (<2% fat): 2 servings, meats, poultry and fish: 3 servings, nuts, seeds and legumes: 2 servings, fats and oils: 3 servings (low in SFA and TC) CG (*n* = 27): control diet (50 to 55% carbohydrate, 15 to 20% protein, and 25 to 30% fat from total E) Dietary components (1600 kcal/day): grains: 9 servings, simple sugar: 4 servings, vegetables: 2 servings, fruits: 2 servings, dairy: 1 serving, meats, poultry and fish: 3 servings, nuts, seeds and legumes: 1 serving, fats and oils: 2 servings Both groups are calorie restricted by BMI (BMI 25 to 29.9 kg/m^2^: 350 kcal, BMI 30 to 39.9 kg/m^2^: 500 kcal) Both groups are isocaloric matched	InG: 32.1 ± 6.0 CG: 31.8 ± 6.2	12 weeks	↓ BW, BMI, WC, HC, LBM, and TFM within InG and CG ↓ BW, BMI, and TFM in InG vs. CG ↔ WC, HC, and LBM between InG and CG ↔ WHR ↓ total T, A4, and FAI within InG and CG ↑ SHBG within InG and CG ↓ A4 in InG vs. CG ↑ SHBG in InG vs. CG ↔ total T and FAI between InG and CG ↑ DPPH within InG and CG ↑ DPPH in InG vs. CG
DASH diet	Foroozanfard et al., 2017 [24]	Iran	Parallel	Overweight and/or obese women with PCOS	InG (*n* = 30): DASH diet (52 to 55% carbohydrate, 16 to 18% protein, and 30% fat from total E; sodium < 2400 mg/day) Dietary components (1800 kcal/day): grains: 7 servings (bread and rice; at least 3 servings from whole grains), simple sugar: 2 servings, vegetables: 5 servings (tomatoes, potatoes, carrots, peas, kale, squash, broccoli, turnip greens, collards, spinach, artichokes and beans), fruits: 6 servings (apricots, bananas, grapes, oranges and juice, tangerines, strawberries, melons, peaches, pineapples, prunes, raisins and grapefruit and juice), dairy: 3 servings (milk, buttermilk, yogurt, or cheese; <2% low fat), meats, poultry and fish: 4 servings (4 servings of lean meats), nuts, seeds and legumes: 2 servings (almonds, mixed nuts, walnuts, sunflower seeds, kidney beans and lentils), and fats and oils: 3 servings (hydrogenated oil, vegetable oils and olive oil) CG (*n* = 30): control diet (52 to 55% carbohydrate, 16 to 18% protein, and 30% fat from total E) Dietary components (1800 kcal/day): grains: 9 servings, simple sugar: 4 servings, vegetables: 4 servings, fruits: 4 servings, dairy: 2 servings, meats, poultry and fish: 4 servings (2 servings of lean meats), nuts, seeds and legumes: 1 serving, and fats and oils: 3 servings Both groups are calorie restricted by BMI (BMI 25 to 27.5 kg/m^2^: 350 kcal; BMI 27.5 to 31 kg/m^2^: 500 kcal; BMI > 31 kg/m^2^: 700 kcal) Both groups are isocaloric matched	InG: 27.1 ± 4.7 CG: 25.6 ± 3.7	12 weeks	↓ BW and BMI in InG vs. CG ↔ FBG within InG and CG ↑ QUICKI within InG ↓ FBI, HOMA-IR, and HOMA-B within InG ↔ FBI, HOMA-IR, HOMA-B, and QUICKI within CG ↓ FBI, HOMA-IR, and HOMA-B in InG vs. CG ↑ QUICKI in InG vs. CG ↔ FBG between InG and CG ↓ FAI within InG ↑ SHBG within InG ↑ AMH within CG ↔ AMH within InG ↔SHBG and FAI within CG ↑ SHBG in InG vs. CG ↓ AMH and FAI in InG vs. CG ↔ total T, FSH, LH, and 17 OH-P ↓ MDA within InG and CG ↑ NO within InG ↔ NO within CG ↓ MDA in InG vs. CG ↑ NO in InG vs. CG
DASH diet	Asemi et al., 2015 [20]	Iran	Parallel	Overweight and/or obese women with PCOS	InG (*n* = 24): DASH diet (52% carbohydrate, 18% protein, and 30% fat from total E; sodium < 2400 mg/day) Dietary components (per day): Grains: 9 servings (≥3 whole grains), simple sugar: 2 servings, vegetables: 5 servings, fruits: 6 servings, low-fat dairy (<2%): 3 servings, lean meat, poultry and fish: 4 servings, nuts, seeds and legumes: 2 servings, and fats and oils: 3 servings CG (*n* = 24): control diet (52% carbohydrate, 18% protein, and 30% fat from total E) Dietary components (per day): Grains: 9 servings, simple sugar: 5 servings, vegetables: 3 servings, fruits: 3 servings, dairy: 2 servings, meat, poultry and fish: 4 servings (lean meat: 2 servings), nuts, seeds and legumes: 1 serving, and fats and oils: 3 servings Both groups were calorie restricted by BMI (BMI 25 to 27.5 kg/m^2^: 350 kcal; BMI 27.5 to 31 kg/m^2^:500 kcal; BMI > 31 kg/m^2^: 700 kcal) Both groups are isocaloric matched	InG: 30.7 ± 6.7 CG: 29.4 ± 6.2	8 weeks	↓ BW, BMI, WC, and HC in InG vs. CG ↓ FBI and HOMA-IR in InG vs. CG ↑ QUICKI in InG vs. CG ↔ FBG and HOMA-B between InG and CG ↓ hs-CRP in InG vs. CG
DASH diet	Asemi et al., 2014 [19]	Iran	Parallel	Overweight and obese women with PCOS	InG (*n* = 24): DASH diet (52% carbohydrate, 18% protein, and 30% fat from total E) Dietary components (per day): Grains: 6 servings (≥3 servings from whole grains), simple sugar: 2 servings, vegetables: 4 servings, fruits: 5 servings, dairy: 3 servings (<2% low fat), lean meats, poultry and fish: 4 servings, nuts, seeds and legumes: 2 servings, and fats and oils: 3 servings CG (*n* = 24): control diet (52% carbohydrate, 18% protein, and 30% fat from total E) Dietary components (per day): Grains: 9 servings, simple sugar: 5 servings, vegetables: 2 servings, fruits: 2 servings, dairy: 2 servings, meats, poultry and fish: 4 servings (lean meats: 2 servings), nuts, seeds and legumes: 1 serving, and fats and oils: 3 servings Both groups are calorie restricted by BMI (BMI 25 to 27.5 kg/m^2^: 350 kcal; BMI 27.5 to 31 kg/m^2^: 500 kcal; BMI > 31 kg/m^2^: 700 kcal) Both groups are isocaloric matched	InG: 22.1 ± 3.2 CG: 24.7 ± 6.0	8 weeks	↓ BW and BMI in InG vs. CG ↔ TC, TG, VLDL-C, HDL-C, LDL-C, and TC/HDL-C ratio ↔ TAC within InG and CG ↑ TAC in InG vs. CG ↔ GSH
Low-carbohydrate diet	Perelman et al., 2017 [57]	USA	Cross-over	Obese women with PCOS	InG (*n* = 6): low-carbohydrate diet [40% carbohydrate, 15% protein, and 45% fat (SFA: ≤7% of E; a ratio of PUFA to MUFA of 1.0) from total E; 200 mg cholesterol and 20 g fiber per day] CG (*n* = 6): control diet [60% carbohydrate, 15% protein, and 25% fat (SFA: ≤7% of E; a ratio of PUFA to MUFA of 1.0) from total E; 200 mg cholesterol and 20 g fiber per day] Both groups are isocaloric matched and maintain BW	30 ± 7	8 weeks (3 weeks for each diet; washout: 2 weeks)	↔ BW between InG and CG ↓ insulin AUC in InG vs. CG ↔ glucose AUC between InG and CG ↓ LDL-C in InG vs. CG ↔ TC, TG and HDL-C between InG and CG
Low-carbohydrate diet	Goss et al., 2014 [55]	USA	Cross-over	Women with PCOS (BMI ≤ 45 kg/m^2^)	InG (*n* = 27): low-carbohydrate diet (41% carbohydrate, 19% protein, and 40% fat from total E; ~50 GI) CG (*n* = 23): control diet (55% carbohydrate, 18% protein, and 27% fat from total E; ~60 GI) Both groups are isocaloric matched and maintain BW	31 ± 5.8	20 weeks (8 weeks for each diet; washout: 4 weeks)	↓ BW, TFM, and thigh SAT within InG and CG ↓ PBF, IAAT, SAAT, and thigh IMAT within InG ↔ LBM within InG ↔ PBF, IAAT, SAAT, and thigh IMAT within CG ↓ LBM within CG ↔ BW, TFM, PBF, LBM, IAAT, SAAT, thigh SAT, and thigh IMAT between InG and CG ↔ thigh PMAT
Low-carbohydrate diet	Gower et al., 2013 [18]	USA	Cross-over	Women with PCOS (BMI ≤ 45 kg/m^2^)	InG (*n* = 27): low-carbohydrate diet (41% carbohydrate, 19% protein, and 40% fat from total E; ~50 GI) Dietary components (1800 kcal/day): carbohydrate 187 g, sugar 79 g, GL 81 g (GL/1000 kcal: 45.90), protein 86 g, SFA 24 g, MUFA 28 g, PUFA 22 g, *n* − 3 FA 1.76 g, and fiber 22 g Dietary components (2500 kcal/day): carbohydrate 261 g, sugar 120 g, GL 114 g (GL/1000 kcal: 46.50), protein 119 g, SFA 34 g, MUFA 40 g, PUFA 29 g, *n*-3 FA 2.45 g, and fiber 31 g CG (*n* = 23): control diet (55% carbohydrate, 18% protein, and 27% fat from total E; ~60 GI) Dietary components (1800 kcal/day): carbohydrate 254 g, sugar 110 g, GL 143 g (GL/1000 kcal: 79.00), protein 84 g, SFA 17 g, MUFA 20 g, PUFA 13 g, *n*-3 FA 0.86 g, and fiber 18 g Dietary components (2500 kcal/day): carbohydrate 350 g, sugar 166 g, GL 192 g (GL/1000 kcal: 76.60), SFA 21 g, MUFA 26 g, PUFA 23 g, *n*-3 FA 1.30 g, and fiber 23 g Both groups are isocaloric matched and maintain BW	31.2 ± 5.8	20 weeks (8 weeks for each diet; washout: 4 weeks)	↔ BW between InG and CG ↓ PhiB, HOMA-IR, FBG, and FBI within InG ↑ PhiD, X0, and Si within InG ↔ PhiB, PhiD, X0, Si, HOMA-IR, FBG, and FBI within CG ↔ PhiS and PhiTOT within InG and CG ↓ HDL-C within InG and CG ↓ TC and LDL-C within InG ↑ TC/HDL-C ratio within CG ↔ TG within InG and CG ↔ TC/HDL-C ratio within InG ↔ TC and LDL-C within CG ↓ total T within InG ↔ total T within CG ↔ FSH, LH, SHBG, and FAI within InG and CG
Low-carbohydrate diet	Douglas et al., 2006 [54]	USA	Cross-over	Overweight and/or obese women with PCOS	InG1 (*n* = 11): low-carbohydrate diet [43% carbohydrate, 15% protein, and 45% total fat (8% SFA, 17% PUFA, and 18% MUFA) from total E; 83 mg cholesterol and 29 g fiber per day] InG2 (*n* = 11): MUFA diet [55% carbohydrate, 15% protein, and 33% fat (7% SFA, 6% PUFA, and 17% MUFA) from total E; 108 mg cholesterol and 24 g fiber per day] CG (*n* = 11): control diet [56% carbohydrate, 16% protein, and 31% fat (7% SFA, 10% PUFA, and 13% MUFA) from total E; 115 mg cholesterol and 27 g fiber per day] All Groups are isocaloric matched and maintain BW (2000 to 2300 kcal/day)	33 ± 6	48 days +3 weeks (16 days for each diets; washout: 3 weeks)	↓ BW in InG1 vs. InG2 ↓ FBI in InG1 vs. CG ↓ AIRg in InG1 vs. InG2 ↔ FBG, Si, and disposition index ↓ TC in InG1 vs. CG ↔ TG, FFA, HDL-C, and LDL-C ↑ A4 in InG2 vs. CG ↔ total T, free T, DHEAS, SHBG, LH, and FSH
Low-carbohydrate diet	Moran et al., 2006 [48]	Australia	Parallel	Overweight and/or obese women with PCOS	InG (phase 1, *n* = 18; phase 2, *n* = 14): low-carbohydrate diet phase 2 (9 to 32 weeks): carbohydrate-restricted BW maintenance diet with low-GI (<120 g carbohydrate/day) CG (phase 1: *n* = 16, phase 2: *n* = 9): control diet phase 2 (9 to 32 weeks): fat-restricted BW maintenance diet with low-GI (<50 g fat/day) Both groups are calorie restricted and consumed same diet in phase 1: phase 1 (0 to 8 weeks): 2 meals with replacements (each meal: ~430 kcal) and a low-fat evening meal (~836 kcal; fruit and vegetables: ≥5 servings/day) Both groups are isocaloric matched	InG: 33.2 ± 4.8 CG: 32.1 ± 5.5	32 weeks (8 weeks: BW loss diet; 24 weeks: BW maintenance diet)	Phase 1 ↓ BW, WC, TFM, FFM, SBP, REE, and RQ within InG and CG ↔ DBP within InG and CG ↔ BW, WC, TFM, FFM, SBP, DBP, REE, and RQ between InG and CG ↓ FBG within InG and CG ↓ FBI and HOMA-IR within InG and CG ↔ glucose AUC within InG and CG ↔ FBG, glucose AUC, FBI, and HOMA-IR between InG and CG ↓ total T, free T, and FAI within InG and CG ↑ SHBG within InG and CG ↔ total T, free T, FAI, and SHBG between InG and CG Phase 2 ↔ BW, WC, TFM, FFM, SBP, DBP, REE, and RQ between InG and CG ↔ FBG, glucose AUC, FBI, and HOMA–IR between InG and CG ↔ total T, free T, FAI, and SHBG between InG and CG
Low-GI diet	Hoover et al., 2021 [56]	USA	Cross-over	Overweight and/or obese women with PCOS	InG (*n* = 27): low-GI diet (41% carbohydrate, 19% protein, and 40% fat from total E; ~50 GI) CG (*n* = 27): control diet (55% carbohydrate, 18% protein, and 27% fat from total E; ~60 GI) Both groups are isocaloric matched and maintain BW	31.2 ± 5.8	20 weeks (8 weeks for each diet; 4 weeks: washout)	↔ BMI between InG and CG ↓ FBG and FBI within InG and CG ↔ FBG and FBI between InG and CG ↔ glucagon ↓ ghrelin and cortisol within InG and CG ↑ GLP-1 within InG and CG ↔ GLP-1, ghrelin, and cortisol between InG and CG ↔ Peptide YY
Low-GI diet	Sordia-Hernández et al., 2016 [49]	Mexico	Parallel	Women with PCOS	InG (*n* = 19): low-GI diet [45% and 50% complex carbohydrate, 15% to 20% protein, and 30% to 40% fat (10% to 15% MUFA, <10% PUFA, and <10% SFA) from total E; 20 to 35 g fiber per day; <45 GI] CG (*n* = 18): control diet [45% and 50% complex carbohydrate, 15% to 20% protein, and 30% to 40% fat (10% to 15% MUFA, <10% PUFA, and <10% SFA) from total E; 20 to 35 g fiber per day; 50 to 75 GI] Both groups are calorie restricted Both groups are isocaloric matched (1200 to 1500 kcal/day)	InG: 26.1 ± 4.2 CG: 26.1 ± 4.7	3 months	↔ BW ↔ cycle 1, cycle 2, and cycle 3 ovulatory cycle between InG and CG ↑ cycle 1 + 2 and cycle 1 + 2+ 3 ovulatory cycle in InG vs. CG
Low-GI diet	Atiomo et al., 2009 [50]	UK	Parallel	Obese women with PCOS	InG (*n* = 6): low-GI diet CG (*n* = 5): control diet Both groups are calorie restricted (600 kcal/day) Both groups are isocaloric matched	InG: 35.3 CG: 36.4	6 months	↔ BW, WC, HC, BMI, SBP, and DBP ↔ FBG and FBI ↔ TC, HDL-C, LDL-C, TG, and HDL-C/LDL-C ratio ↔ ED, LH, SHBG, and T ↔ endometrial thickness
low-GI plus high-protein diet	Mehrabani et al., 2012 [16]	Iran	Parallel	Overweight and/or obese women with PCOS	InG (*n* = 23): low-GI plus high-protein diet (40% carbohydrate, 30% protein, and 30% fat from total E; <20 GL foods) CG (*n* = 26): control diet (55% carbohydrate, 15% protein, and 30% fat from total E) Both groups were calorie restricted by BMI (BMI 21 to 22 kg/m^2^: 500 to 1000 kcal/day) Both groups are isocaloric matched	InG: 30.5 ± 6.4 CG: 28.5 ± 5.2	12 weeks	↓ BW within InG and CG ↓ WC and HC in InG vs. CG ↔ BW, PBF, percent of LBM, and suprailiac skinfold between InG and CG ↓ FBI and HOMA-IR within InG ↔ FBI and HOMA-IR within CG ↓ adiponectin within InG and CG ↓ FBI and HOMA-IR in InG vs. CG ↔ adiponectin between InG and CG ↓ LDL-C within InG and CG ↔ LDL-C between InG and CG ↔ TG, TC, and HDL-C ↑ SHBG within InG and CG ↓ total T, DHEAS, and FAI within InG and CG ↔ SHBG, total T, DHEAS, and FAI between InG and CG ↔ A4, FSH, and LH ↓ hs-CRP within InG ↔ hs-CRP between InG and CG ↓ TNF-α within InG and CG ↔ TNF-α between InG and CG ↔ IL-6
High-protein diet	Sørensen et al., 2012 [17]	Denmark	Parallel	Women with PCOS	InG (*n* = 14): high-protein diet (<30% carbohydrate, >40% protein, and 30% fat from total E) CG (*n* = 13): control diet (>55% carbohydrate, <15% protein, and 30% fat from total E) Both groups are not calorie restricted Both groups are isocaloric matched (mean difference = ~136 kcal; 95% CI ~−134 to ~406 kcal; *p* = 0.16)	InG: 27.7 ± 5.5 CG: 28.4 ± 5.8	6 months	↓ BW, BMI, TFM, and WC within InG and CG ↓ BW, BMI, TFM, and WC in InG vs. CG ↔ LBM, HC, and WHR ↓ FBG in InG vs. CG ↔ C-peptide between InG and CG ↔ TG, HDL-C, LDL-C, TC, TC/HDL-C ratio, and TG/HDL-C ratio between InG and CG ↔ total T, free T, and SHBG between InG and CG
High-protein diet	Moran et al., 2010 [15]	Australia	Parallel	Overweight and/or obese women with PCOS	InG (*n* = 14): high-protein diet (43% carbohydrate, 27% protein, and 28% fat from total E) CG (*n* = 14): control diet (57% carbohydrate, 16% protein, and 27% fat from total E) Both groups are calorie restricted Both groups are isocaloric matched (~1434 kcal/day)	32.8 ± 4.5	16 weeks (12 weeks: calorie-restricted diet, 4 weeks: calorie balance diet)	↓ BW, BMI, and MAP within InG and CG ↔ BW, BMI, and MAP between InG and CG ↓ FBI and insulin AUC within InG and CG ↔ FBG and glucose AUC between InG and CG ↑ LAE and SAE within InG and CG ↔ LAE and SAE between InG and CG ↓ TG and FFA within InG and CG ↔ TG and FFA between InG and CG ↔ TG AUC and FFA AUC ↔ FAI ↓ hs-CRP within InG and CG ↔ hs-CRP between InG and CG
High-protein diet	Kasim-Karakas et al., 2009 [52]	USA	Parallel	Overweight and/or obese women with PCOS	InG (*n* = 11): 240 kcal whey protein + calorie-restricted diet (39.5% carbohydrate, 33.7% protein, and 26.2% fat from total E) CG (*n* = 13): 240 kcal simple sugars + calorie-restricted diet (56.7% carbohydrate, 16.6% protein, and 25.9% fat from total E) Both groups are calorie restricted (450 kcal/day) Both groups are isocaloric matched	18 to 45	2 months	↓ BMI within InG and CG ↓ BW and TFM within InG ↔ BW and TFM within CG ↓ BW, BMI, and TFM in InG vs. CG ↔ LBM ↓ leptin and adiponectin within InG ↔ leptin and adiponectin between InG and CG ↔ FBG, FBI, HbA1c, and HOMA-IR ↓ TC, HDL-C, and apo B within InG ↔ TC, HDL-C, and apo B within CG ↓ TC, HDL-C, and apo B in InG vs. CG ↔ TG ↔ T, SHBG, FAI, and DHEAS ↔ hs-CRP
High-protein diet	Stamets et al., 2004 [14]	USA	Parallel	Overweight and/or obese women with PCO	InG (*n* = 13): high-protein diet (40% carbohydrate, 30% protein, and 30% fat from total E) CG (*n* = 13): control diet (55% carbohydrate, 15% protein, and 30% fat from total E) Both groups are calorie restricted (1000 kcal/day) and consume multivitamin/mineral supplements Both groups are isocaloric matched	InG: 29 ± 4 CG: 26 ± 4	4 weeks	↓ BW within InG and CG ↓ WC within InG ↔ BW, WC, HC, WHR, SBP, and DBP between InG and CG ↓ fasting leptin and leptin AUC within InG ↔ FBI, insulin AUC and FBG/FBI ratio within InG ↔ insulin AUC, glucose AUC, FBG/FBI ratio, fasting leptin, and leptin AUC between InG and CG ↓ TC and LDL-C within InG ↔ TC, HDL-C, LDL-C, and TG between InG and CG ↑ DHEAS within InG ↓ total T and bioavailable T within InG ↔ DHEAS, total T, bioavailable T, LH, and FSH between InG and CG
High-protein diet	Moran et al., 2003 [13]	Australia	Parallel	Overweight and/or obese women with PCOS	InG (*n* = 14): high-protein diet (40% carbohydrate, 30% protein, and 30% fat from total E) CG (*n* = 14): control diet (55% carbohydrate, 15% protein, and 30% fat from total E) Both groups are calorie restricted and isocaloric matched (~1433 kcal/day)	InG: 32 ± 1.2 CG: 33 ± 1.2	16 weeks (12 weeks: calorie-restricted diet; 4 weeks: BW maintenance)	12 weeks (vs. baseline) ↓ FBI and HOMA-IR within InG and CG ↔ FBI and HOMA-IR between InG and CG ↔ FBG ↓ TC, LDL-C, TG, and TC/HDL-C ratio within InG and CG ↔ HDL-C within InG ↓ HDL-C within CG ↓ HDL-C in CG vs. InG ↓ TC/HDL-C ratio in InG vs. CG ↔ TC, LDL-C, and TG between InG and CG ↑ SHBG within InG and CG ↓ total T and FAI within InG and CG ↔ total T, FAI, and SHBG between InG and CG 16 weeks (vs. baseline) ↔ BW within InG and CG ↓ TFM, LBM, and abdominal fat mass within InG and CG ↔ BW, TFM, LBM, and abdominal fat mass between InG and CG ↓ FBI and HOMA-IR within InG and CG ↑ glucose AUC within CG ↔ FBI, HOMA-IR, and insulin AUC between InG and CG ↔ FBG ↓ TC, LDL-C, and TG within InG and CG ↓ TC/HDL-C in InG vs. CG ↑ HDL-C in InG vs. CG

AIRg, acute insulin response to glucose; AMH, anti-Müllerian hormone; apo B, apoprotein B; AUC, area under the curve; A4, androstenedione; BMI, body mass index; BW, body weight; CG, control group; CI, confidence interval; CO, crossover; DASH, Dietary Approaches to Stop Hypertension; DBP, diastolic blood pressure; DHEAS, dehydroepiandrosterone sulfate; DPPH, 2,2′-diphenyl-1-picryylhydrazyl; E, energy; ED, estradiol; FAI, free androgen index; FBG, fasting blood glucose; FBI, fasting blood insulin; FFA, free fatty acids; FFM, fat-free mass; FSH, follicle-stimulating hormone; GSH, total glutathione; GI, glycemic index; GL, glycemic load; GLP-1, glucagon-like peptide-1; HbA1c, hemoglobin A1c; HC, hip circumference; HDL-C, high-density lipoprotein cholesterol; HOMA-B, homeostasis model assessment of beta-cell function; HOMA-IR, homoeostasis model of assessment-estimated insulin resistance; hs-CRP, high sensitivity C-reactive protein; IAAT, intra-abdominal adipose tissue; IMAT, intermuscular adipose tissue; InG, intervention group; LAE, proximal arterial compliance; LBM, lean body mass; LDL-C, low-density lipoprotein cholesterol; LH, luteinizing hormone; MAP, mean arterial pressure; MDA, malondialdehyde; mo, months; MUFA, monounsaturated fatty acids; *n*, number; NO, nitric oxide; *n*-3 FA, omega-3 fatty acid; P, parallel; PBF, percent of body fat mass; PCOS, polycystic ovary syndrome; PhiB, basal beta-cell response to glucose; PhiD, dynamic beta-cell response to glucose; PhiS, static beta-cell response to glucose; PhiTOT, total beta-cell response to glucose; PMAT, perimuscular adipose tissue; PUFA, polyunsaturated fatty acid; QUICKI, quantitative insulin sensitivity check index; RQ, respiratory quotient; SAAT, subcutaneous abdominal adipose tissue; SAE, small resistance arterial compliance; SAT, subcutaneous adipose tissue; SBP, systolic blood pressure; SFA, Saturated fatty acid; SHBG, sex hormone binding globulin; Si, insulin sensitivity; T, testosterone; TAC, total antioxidant capacity; TC, total cholesterol; TFM, total fat mass; TG, triglyceride; TNF-α, tumor necrosis factor alpha; UK, United Kingdom; USA, United States of America; VLDL-C, very-low-density lipoprotein cholesterol; WC, waist circumference; WHR, waist-to-hip ratio; wks, weeks; X0, volume of insulin secreted in first phase; 17 OH-P, 17-hydroxyprogesterone; ↑, significant increase in outcome; ↓, significant decrease in outcome; ↔, no significant effect.

**Table 2 nutrients-17-00674-t002:** Effects of dietary patterns in individuals with polycystic ovary syndrome in the dietary RCTs differing in calories.

Dietary Pattern	Reference	Region	Study Design	Subject Health Status	Dietary Intervention	Mean Age (Years)	Study Period	Outcomes
KD	Cincione et al., 2023 [47]	Italy	Parallel	Overweight and/or obese women with PCOS	InG (*n* = 73): mixed KD with caloric-restriction (~600 kcal/day) [≤30 g carbohydrate, 1.1 to 1.2 g protein/ideal BW (isolated whey protein powder), 30 g lipid (10 g EVOO) per day;multivitamin and multi-mineral supplement] CG (*n* = 71): MED diet with calorie restriction (500 kcal restriction/day) [55% carbohydrate (whole wheat), 20% protein (fish and legumes), and 25% fat (PUFA from olive oil, almonds, and pistachios) from total E] Both groups are not isocaloric matched	InG: 33.4 ± 5.7 CG: 33.6 ± 4.9	45 days	↓ BW, BMI, WC, HC, WHR, TFM, FFM, REE, BM, and TBW within InG and CG ↓ BW, BMI, WC, HC, WHR, TFM, FFM, REE, BM, and TBW in InG vs. CG ↓ FBG, FBI, HOMA-IR, and C-peptide within InG and CG ↑ albumin within InG and CG ↓ FBG, FBI, HOMA-IR, and C-peptide in InG vs. CG ↑ albumin in InG vs. CG ↓ total T, free T, LH, and LH/FSH ratio within InG and CG ↑ FSH and SHBG within InG and CG ↓ total T, free T, and LH in InG vs. CG ↑ FSH and SHBG in InG vs. CG ↔ LH/FSH ratio between InG and CG ↑ regular menstrual cycle in InG vs. CG
High-protein diet	Dou et al., 2024 [51]	China	Parallel	Overweight and/or obese women without PCOS	InG1 (*n* = 30): balanced diet + high-protein + high-fiber (10 g fiber/day) InG2 (*n* = 30): balanced diet + high-protein (40 g whey protein; daily protein intake of 1.5 to 2 kg/day) CG (*n* = 22): balanced diet (55 to 60% carbohydrate, 0.8 to 1.2 g/kg/day protein and 25 to 30% fat from total E) Balanced diet is calorie restricted Balanced dietary components (1000 to 1200 kcal/day): Breakfast: 0.5 to 1 fist-sized cooked staple food (0.5 to 1 tael raw rice and noodles), 250 mL low-fat milk, 1 egg and 100 g leafy vegetables; Lunch: 0.5 to 1 fist-sized cooked staple food (0.5 to 1 tael raw rice and noodles), 100 g low-fat lean meat and leafy vegetables; Dinner: 0.5 to 1 fist-sized cooked staple food (0.5 to 1 tael raw rice and noodles), 100 g bean products, and leafy vegetables; The morning meal and afternoon meal: 100 g fruit, respectively	InG1: 31.3 ± 5.8 InG2: 32.1 ± 5.8 CG: 31.5 ± 2.7	8 weeks	↓ BW, BMI, PBF, and VFA within InG1 and InG2 and CG ↓ FFM within CG ↔ FFM within InG1 and InG2 ↔ FFM index within InG1 and InG2 and CG ↓ BW and PBF in InG1 and InG2 vs. CG ↓ BMI and VFA in InG1 vs. InG2 and CG ↔ FFM and FFM index between InG1 and InG2 and CG ↓ HOMA-IR within InG1 and InG2 and CG ↔ HOMA-IR between InG1 and InG2 and CG ↓ MDA within InG1 and InG2 and CG ↓ MDA in InG1 and InG2 vs. CG ↑ SOD within InG1 and InG2 and CG ↑ SOD in InG1 and InG2 vs. CG
Very low-calorie diet	Deshmukh et al., 2023 [53]	UK	Parallel	Obese women with PCOS	InG (*n* = 11): very low-calorie diet Diet intervention (per day): 800 kcal meal replacement (each meal: 200 kcal; 21 g carbohydrate, 15 g protein, and 3 to 4 g fat) Diet reintroduction: calorie increase (200 kcal/2 weeks) until ~1600 kcal/day CG (*n* = 11): low-calorie diet (600 kcal restriction/day)	InG: 27.7 ± 3.8 CG: 28.1 ± 5.6	16 weeks (8 weeks: intervention; 8 weeks: reintroduction)	8 weeks ↓ BW, BMI, WC, WHR, TFM, trunk fat, LBM, and FFM within InG ↓ BW, BMI, and TFM within CG ↔ WC, WHR, trunk fat, LBM, and FFM within CG ↓ BW, BMI, WC, WHR, total fat, trunk fat, LBM, and FFM in InG vs. CG ↔ BMC and BMD ↓ FBG within InG ↔ FBG within CG ↓ FBG in InG vs. CG ↔ 2 h glucose OGTT and HbA1c ↓ TC within InG and CG ↔ TC between InG and CG ↔ TG ↓ FAI within InG ↔ FAI and SHBG within CG ↑ SHBG within InG ↑ SHBG in InG vs. CG ↔ FAI between InG and CG ↔ total T, DHEAS, A4, LH, and FSH ↔ CRP, ALT, and AST 16 weeks ↓ BW in InG vs. CG ↔ FAI between InG and CG

ALT, alanine aminotransferase; AST, aspartate aminotransferase; A4, androstenedione; BM, basal metabolism; BMC, body composition content; BMD, bone mineral density; BMI, body mass index; BW, body weight; CG, control group; CRP, C-reactive protein; DHEAS, dehydroepiandrosterone sulfate; E, energy; EVOO, extra virgin olive oil; FAI, free androgen index; FBG, fasting blood glucose; FBI, fasting blood insulin; FFM, fat-free mass; FSH, follicle-stimulating hormone; HbA1c, hemoglobin A1c; HC, hip circumference; HOMA-IR, homoeostasis model of assessment-estimated insulin resistance; InG, intervention group; KD, ketogenic diet; LBM, lean body mass; LH, luteinizing hormone; MDA, malondialdehyde; MED, Mediterranean; *n*, number; OGTT, oral glucose tolerance test; P, parallel; PBF, percent of body fat; PCOS, polycystic ovary syndrome; PUFA, polyunsaturated fatty acid; REE, resting energy expenditure; SHBG, sex hormone binding globulin; SOD, superoxide dismutase; T, testosterone; TC, total cholesterol; TFM, total fat mass; TG, triglyceride; TBW, total body water; UK, United Kingdom; VFA, visceral fat area; WC, waist circumference; WHR, waist-to-hip ratio; wks, weeks; ↑, significant increase in outcome; ↓, significant decrease in outcome; ↔, no significant effect.

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
