# Peer review of "The Influence of Dietary Patterns on Polycystic Ovary Syndrome Management in Women: A Review of Randomized Controlled Trials with and Without an Isocaloric Dietary Design"

_nutrients, 2025, doi:10.3390/nu17040674_

Round 1

Reviewer 1 Report

Comments and Suggestions for Authors

The article deals with the effects of dietary patterns on the management of polycystic ovary syndrome (PCOS) in women. The topic is relevant as PCOS is one of the most common endocrine metabolic disorders, the treatment of which is largely based on lifestyle modification, including diet. 

The analysis is based on a review of randomised clinical trials (RCTs), which are the gold standard in research.

Consideration of different dietary approaches, including DASH, low-carbohydrate, low-glycemic and high-protein diets.

Comprehensive assessment of metabolic and hormonal parameters, allowing a more holistic view of the topic.

The authors point out that only a limited number of RCTs are available (21 studies), which limits the ability to draw firm conclusions.

The neutral effects of some diets suggest that there was insufficient control for external variables in the studies analysed.

The conclusions suggest that the calorie-restricted version of the DASH diet is the most effective for improving metabolic parameters and reducing body weight in women with PCOS.

The low glycemic and high protein diets showed only neutral effects on glycemic control, lipids and gonadal parameters.

The authors emphasise that further studies are needed, especially those that consider isocaloric dietary patterns.

Transparent literature search strategy in the PubMed/MEDLINE database.

Inclusion of a wide range of key terms, which increases the chances of a comprehensive collection of studies.

Clear inclusion and exclusion criteria for studies, e.g. exclusion of non-clinical, observational studies without English-language full text.

Lack of detailed discussion of the methodological quality of the studies analysed (e.g. use of instruments to assess risk of bias).

Insufficient discussion of the diversity of the population of women studied (e.g. ethnic or regional differences that may influence the results).

The authors focused on comparing the intervention and control groups on isocaloric and different caloric diets, which is important from a methodological perspective.

The results are presented in tables to facilitate comparison, but the statistical analyses of the cited studies are not discussed in detail.

The article is well organised, with a clear division into sections such as introduction, methods, results and discussion.

The PRISMA diagram correctly illustrates the study selection process, demonstrating the reliability of the review methodology.

In the Discussion, it would be useful to add a more detailed comparison of the effects of the different diets and their potential mechanisms of action in relation to PCOS.

The conclusions could be more precise and relate to specific clinical recommendations, e.g. which diets can be recommended depending on the specific therapeutic goal (e.g. weight loss, improvement of hormonal function).

The language of the article is clear and precise, so that even the more technical parts of the text are easy to understand.

Standard scientific terminology is used, which is important in this type of publication.

The flow of the text could be improved in the abstract to summarise the main results more clearly.

Several repetitions in the results section on the neutral effects of diets could be shortened or summarised to improve readability.

Author Response

< Responses to the “reviewer 1” comments >

The article deals with the effects of dietary patterns on the management of polycystic ovary syndrome (PCOS) in women. The topic is relevant as PCOS is one of the most common endocrine metabolic disorders, the treatment of which is largely based on lifestyle modification, including diet.

The analysis is based on a review of randomised clinical trials (RCTs), which are the gold standard in research.

Consideration of different dietary approaches, including DASH, low-carbohydrate, low-glycemic and high-protein diets.

Comprehensive assessment of metabolic and hormonal parameters, allowing a more holistic view of the topic.

The authors point out that only a limited number of RCTs are available (21 studies), which limits the ability to draw firm conclusions.

The neutral effects of some diets suggest that there was insufficient control for external variables in the studies analysed.

The conclusions suggest that the calorie-restricted version of the DASH diet is the most effective for improving metabolic parameters and reducing body weight in women with PCOS.

The low glycemic and high protein diets showed only neutral effects on glycemic control, lipids and gonadal parameters.

The authors emphasise that further studies are needed, especially those that consider isocaloric dietary patterns.

Transparent literature search strategy in the PubMed/MEDLINE database.

Inclusion of a wide range of key terms, which increases the chances of a comprehensive collection of studies.

Clear inclusion and exclusion criteria for studies, e.g. exclusion of non-clinical, observational studies without English-language full text.

Lack of detailed discussion of the methodological quality of the studies analysed (e.g. use of instruments to assess risk of bias).

→ Thank you for your comment. It has been addressed in lines 984-985.
In lines 984-985

“No risk of bias methods was used to assess the included RCTs.”

Insufficient discussion of the diversity of the population of women studied (e.g. ethnic or regional differences that may influence the results).

→ Thank you for your comment. It has been addressed in lines 171-172.

In lines 171-172

“The RCTs were conducted in Iran [16,19,20,23,24], China [51], Mexico [49], USA [14,18,52,54-57], Italy [47], Australia [13,15,48], the United Kingdom [50,53], and Denmark [17].”

The authors focused on comparing the intervention and control groups on isocaloric and different caloric diets, which is important from a methodological perspective.

The results are presented in tables to facilitate comparison, but the statistical analyses of the cited studies are not discussed in detail.

→ Thank you for your comment. We changed the descriptions of the shorthand symbols in the footnotes below the table to present the statistical significance of the results in lines 194-195 (Table 1) and 205 (Table 2).

In lines 194-195 and 205

 “↑, significant increase in outcome; ↓, significant decrease in outcome; ↔, no significant effect”

The article is well organised, with a clear division into sections such as introduction, methods, results and discussion.

The PRISMA diagram correctly illustrates the study selection process, demonstrating the reliability of the review methodology.

In the Discussion, it would be useful to add a more detailed comparison of the effects of the different diets and their potential mechanisms of action in relation to PCOS.

→ Thank you for your comment. It has been addressed in lines 663-669, 693-700, 749-755, 841-846, and 867-870.

In lines 663-669

“There is some evidence that the DASH diet improves the metabolic characteristics of women with PCOS [19,20,23,24]. Clinical trials on women with PCOS showed that weight loss from the DASH diet enhanced Si [20,24]. The weight loss reduced leptin [79] and increased concentrations of insulin-like growth factor-binding protein-1 [80] and adiponectin [81]. The excess adiponectin level stimulated the down regulation of adipose tissue [81]. In addition, the dietary antioxidants, mainly from fruits and vegetable rich in the DASH diet, alleviated inflammation and oxidative stress [62,82].”

[79] Herrick, J.E.; Panza, G.S.; Gollie, J.M. Leptin, Leptin Soluble Receptor, and the Free Leptin Index following a Diet and Physical Activity Lifestyle Intervention in Obese Males and Females. J Obes 2016, 2016, 8375828, doi:10.1155/2016/8375828.

[80] Belobrajdic, D.P.; Frystyk, J.; Jeyaratnaganthan, N.; Espelund, U.; Flyvbjerg, A.; Clifton, P.M.; Noakes, M. Moderate energy restriction-induced weight loss affects circulating IGF levels independent of dietary composition. Eur J Endocrinol 2010, 162, 1075-1082, doi:10.1530/eje-10-0062.

[81] Nigro, E.; Scudiero, O.; Monaco, M.L.; Palmieri, A.; Mazzarella, G.; Costagliola, C.; Bianco, A.; Daniele, A. New insight into adiponectin role in obesity and obesity-related diseases. Biomed Res Int 2014, 2014, 658913, doi:10.1155/2014/658913.

[82] Soltani, S.; Chitsazi, M.J.; Salehi-Abargouei, A. The effect of dietary approaches to stop hypertension (DASH) on serum inflammatory markers: A systematic review and meta-analysis of randomized trials. Clin Nutr 2018, 37, 542-550, doi:10.1016/j.clnu.2017.02.018.

In lines 693-700

“Although the KD has been demonstrated to have a favorable impact on anthropometric parameters, glucose control, and gonadal parameters in women with PCOS, the mechanisms underlying its beneficial effects remain to be elucidated. The KD stimulates nutritional ketosis through the restriction of carbohydrates, thereby increasing lipolysis and enhancing metabolic efficiency [93]. It has been postulated that the restriction of carbohydrates improves Si, thereby enhancing endocrine function [93]. Moreover, the insulin level has been associated with androgen levels and SHBG synthesis in women with PCOS [94].”

[93] Khalid, K.; Apparow, S.; Mushaddik, I.L.; Anuar, A.; Rizvi, S.A.A.; Habib, A. Effects of Ketogenic Diet on Reproductive Hormones in Women With Polycystic Ovary Syndrome. J Endocr Soc 2023, 7, bvad112, doi:10.1210/jendso/bvad112.

[94] Zhu, H.; Bi, D.; Zhang, Y.; Kong, C.; Du, J.; Wu, X.; Wei, Q.; Qin, H. Ketogenic diet for human diseases: the underlying mechanisms and potential for clinical implementations. Signal Transduct Target Ther 2022, 7, 11, doi:10.1038/s41392-021-00831-w.

In lines 749-755

“A low-carbohydrate diet has been shown to bring beneficial effects on weight loss and ovarian function, reducing concentrations on FBG, FBI, insulin-like growth factor-1 and insulin-like growth factor-binding protein-1 [98]. In addition, a low-carbohydrate diet was found to decrease TG and increase HDL-C compared with the high-carbohydrate diet [99]. Moreover, a low-carbohydrate diet restores the balance of inositol metabolism, which can increase Si [100]. This can help improve insulin resistance effectively, lower androgen levels, and enhance menstrual cycle in women with PCOS [101].”

[98] Zhang, X.; Zheng, Y.; Guo, Y.; Lai, Z. The Effect of Low Carbohydrate Diet on Polycystic Ovary Syndrome: A Meta-Analysis of Randomized Controlled Trials. Int J Endocrinol 2019, 2019, 4386401, doi:10.1155/2019/4386401.

[99] Tay, J.; Luscombe-Marsh, N.D.; Thompson, C.H.; Noakes, M.; Buckley, J.D.; Wittert, G.A.; Yancy, W.S., Jr.; Brinkworth, G.D. Comparison of low- and high-carbohydrate diets for type 2 diabetes management: a randomized trial. Am J Clin Nutr 2015, 102, 780-790, doi:10.3945/ajcn.115.112581.

[100] Laganà, A.S.; Garzon, S.; Casarin, J.; Franchi, M.; Ghezzi, F. Inositol in Polycystic Ovary Syndrome: Restoring Fertility through a Pathophysiology-Based Approach. Trends Endocrinol Metab 2018, 29, 768-780, doi:10.1016/j.tem.2018.09.001.

[101] Xu, Y.; Qiao, J. Association of Insulin Resistance and Elevated Androgen Levels with Polycystic Ovarian Syndrome (PCOS): A Review of Literature. J Healthc Eng 2022, 2022, 9240569, doi:10.1155/2022/9240569.

In lines 841-846

“Previous studies have hypothesized that a low-GI diet could play a significant role in glucose homeostasis, thereby alleviated the anthropometric and metabolic profile of women with PCOS [108]. The high-GI food-induced insulin elevation increased the lipogenesis in the adipose tissue [73]. A low-GI diet led to a gradual increment of postprandial glycemia, lowering insulin resistance [74] and the risk of PCOS exacerbation [109]. Short-term use of a low-GI diet could slightly improve Si in women with PCOS [110].”

[108] Manta, A.; Paschou, S.A.; Isari, G.; Mavroeidi, I.; Kalantaridou, S.; Peppa, M. Glycemic Index and Glycemic Load Estimates in the Dietary Approach of Polycystic Ovary Syndrome. Nutrients 2023, 15, doi:10.3390/nu15153483.

[109] Sadeghi, H.M.; Adeli, I.; Calina, D.; Docea, A.O.; Mousavi, T.; Daniali, M.; Nikfar, S.; Tsatsakis, A.; Abdollahi, M. Polycystic Ovary Syndrome: A Comprehensive Review of Pathogenesis, Management, and Drug Repurposing. Int J Mol Sci 2022, 23, doi:10.3390/ijms23020583.

[110] Barr, S.; Reeves, S.; Sharp, K.; Jeanes, Y.M. An isocaloric low glycemic index diet improves insulin sensitivity in women with polycystic ovary syndrome. J Acad Nutr Diet 2013, 113, 1523-1531, doi:10.1016/j.jand.2013.06.347.

In lines 867-870

“A high-protein diet has been suggested as effective treatment for weight loss [111]. In addition, a high-protein diet was associated with the reductions in HOMA-IR, LDL-C, TC, and TG in individuals with T2DM [112]. Furthermore, a high-protein diet significantly decreased FBI and HOMA-IR in women with PCOS [111].”

[111] Wang, F.; Dou, P.; Wei, W.; Liu, P.J. Effects of high-protein diets on the cardiometabolic factors and reproductive hormones of women with polycystic ovary syndrome: a systematic review and meta-analysis. Nutr Diabetes 2024, 14, 6, doi:10.1038/s41387-024-00263-9.

[112] Yu, Z.; Nan, F.; Wang, L.Y.; Jiang, H.; Chen, W.; Jiang, Y. Effects of high-protein diet on glycemic control, insulin resistance and blood pressure in type 2 diabetes: A systematic review and meta-analysis of randomized controlled trials. Clin Nutr 2020, 39, 1724-1734, doi:10.1016/j.clnu.2019.08.008.

The conclusions could be more precise and relate to specific clinical recommendations, e.g. which diets can be recommended depending on the specific therapeutic goal (e.g. weight loss, improvement of hormonal function).

→ Thank you for your comment. It has been changed in lines 1002-1004.

In lines 1002-1004

“Especially, adherence to the DASH dietary pattern has been suggested as a potential treatment approach for weight loss and glucose control.”

The language of the article is clear and precise, so that even the more technical parts of the text are easy to understand.

Standard scientific terminology is used, which is important in this type of publication.

The flow of the text could be improved in the abstract to summarise the main results more clearly.

→ Thank you for your comment. We have improved the abstract to summarise the main results more clearly.

In lines 25-33

“This review found beneficial effects of the calorie-restricted Dietary Approaches to Stop Hypertension (DASH) diet on weight loss and glucose control in women with PCOS in 4 RCTs with an isocaloric dietary design. The calorie-restricted low-glycemic index (GI) diets from 3 RCTs and high-protein diets from 4 RCTs with an isocaloric dietary design showed no significant differences in anthropometric parameters, glucose control, lipids, and gonadal parameters compared with the control diet in women with PCOS. Non-calorie-restricted low-carbohydrate diets from 4 RCTs with an isocaloric dietary design showed similar results to the calorie-restricted low-GI diets and high-protein diets.”

Several repetitions in the results section on the neutral effects of diets could be shortened or summarised to improve readability.

→ Thank you for your comment. It has been changed in lines 397-400, 498-500, 596-597, and 749-751.

In lines 397-400

“The finding showed that a calorie-restricted low-GI diet was not associated with changes in BW [16,49,50], WC [16,50], HC [16,50], FBI [16,50], and lipids (TC, LDL-C, HDL-C, and TG) [16,50], gonadal parameters (total T, SHBG, and LH) [16,50].”

In lines 498-500

“In conclusion, no significant differences in anthropometric parameters, glucose control, and gonadal parameters were observed between the high-protein diet and the control diet.”

In lines 596-597

“The low-carbohydrate diet was found to have no significant association with BW reduction, glucose control, and lipids in women with PCOS.”

In lines 749-751

“With regard to gonadal parameters, a low-carbohydrate diet showed no significant differences in total T [48,54], free T [48,54], SHBG [48,54], DHEAS [54], FAI [48], LH [54], and FSH [54] compared with the control diet in women with PCOS.”

Reviewer 2 Report

Comments and Suggestions for Authors

Dear authors,

The study, The Influence of Dietary Patterns on Polycystic Ovary Syndrome Management in Women: A Review of Randomized Controlled Trials with and without an Isocaloric Dietary Design, is quite interesting and relevant to the field. PCOS is a condition that affects many women around the world and requires better understanding.

The study is clear and presents interesting results. I think the locations where the studies were carried out should be added to the tables. It should also be included whether the studies considered the use of medication or even physical exercise in the population studied.

They must also present a topic with the limitations and perspectives of the study.

Author Response

< Responses to the “reviewer 2” comments >

Dear authors,

The study, The Influence of Dietary Patterns on Polycystic Ovary Syndrome Management in Women: A Review of Randomized Controlled Trials with and without an Isocaloric Dietary Design, is quite interesting and relevant to the field. PCOS is a condition that affects many women around the world and requires better understanding.

The study is clear and presents interesting results. I think the locations where the studies were carried out should be added to the tables.

→ Thank you for your comment. It has been addressed in lines 171-172 and Table 1 and Table 2 (lines 178 and 196).

In lines 171-172

“The RCTs were conducted in Iran [16,19,20,23,24], China [51], Mexico [49], USA [14,18,52,54-57], Italy [47], Australia [13,15,48], UK [50,53], and Denmark [17].”

In lines 178 and 196 (Table 1 and Table 2)

“Region”

It should also be included whether the studies considered the use of medication or even physical exercise in the population studied.

→ Thank you for your comment. This review excluded RCTs that included physical exercises, medications, extracts, or supplements in the intervention design to examine the effect of dietary patterns on PCOS in women. It has been addressed in lines 110-112.

In lines 110-112

“With regard to unrelated intervention design, this study excluded studies if the intervention included physical exercises, medications, extracts, or supplements.”

They must also present a topic with the limitations and perspectives of the study.

→ Thank you for your comment. It has been addressed in lines 1004-1006.

In lines 1004-1006

“However, further intervention studies that conduct isocaloric dietary design or recruit subjects in Asian countries are necessary to verify the role of dietary patterns on PCOS in women.”

Round 2

Reviewer 2 Report

Comments and Suggestions for Authors

Dear authors,

Thanks for your reply. I  don´t have more questions. 

Congrats.

Author Response

Thank you.